# Width Independent Bounds for the Local Lipschitz Constant of Deep Neural Networks at Random Initialization and after Lazy Training

**Apostolos Evangelidis** [1 2]   **Felix Krahmer** [1 2 3]

## Abstract

A plethora of recent works has shown that for wide, overparameterized neural networks, training with Stochastic Gradient Descent (SGD) often leads to interpolation of the training data without sacrificing generalization performance. A key parameter that is not only closely connected to generalization properties, but is also closely tied to other desiderata such as robustness and resistance to adversarial perturbations is the Lipschitz constant of the neural network. While empirically, the Lipschitz constant has been shown not to increase with network width, theoretical findings only provide bounds with logarithmic growth in the width and only for the random initialization of ReLU-networks. In this work, we close this gap for neural networks with smooth activations by showing that, both at random initialization and throughout lazy training, the local Lipschitz constant of deep neural networks does not increase with network width. More precisely, we establish novel non-asymptotic (finite width) upper bounds and corroborate them by numerical experiments.

## 1. Introduction

Modern learning theory faces a fundamental discrepancy: neural networks trained in heavily overparameterized regimes often generalize well (Peleg & Hein, 2024), even though classical statistical arguments predict severe overfitting (Vapnik, 2013). An analogous contradiction appears in the context of adversarial robustness. While empirical studies often report that larger, overparameterized models

[1]Department of Mathematics, Technical University of Munich, Munich, Germany [2]Munich Center for Machine Learning, Munich, Germany [3]Department of Electrical Engineering and Information Technology, Technical University of Darmstadt, Darmstadt, Germany. Correspondence to: Apostolos Evangelidis <apostolos.evangelidis@tum.de>, Felix Krahmer <felix.krahmer@tu-darmstadt.de>.

*Proceedings of the 43rd International Conference on Machine Learning*, Seoul, South Korea. PMLR 306, 2026. Copyright 2026 by the author(s).

are more robust (Madry et al., 2018), theoretical results suggest that increasing model capacity should instead have the opposite effect (Hassani & Javanmard, 2024). This apparent contradiction is often explained by the implicit bias induced by the training algorithm, architecture, and initialization, which favors solutions with better generalization and robustness properties despite overparameterization.

A key quantity of any predictive model is its *Lipschitz constant*, as it quantifies the model's sensitivity to small input perturbations. More precisely, the Lipschitz constant acts as a global bound on how much the output can change relative to a given change in the input. Notably, prior works have established clear correlations between its value and a model's sensitivity to adversarial perturbations (Yang et al., 2020) as well as its generalization behavior (Bartlett et al., 2017). It is therefore natural to study the Lipschitz constant of neural networks, as it offers a principled lens through which to better understand the interplay between overparameterization, generalization, and adversarial robustness.

However, exactly computing the Lipschitz constant of a neural network is NP-hard (Virmaux & Scaman, 2018; Jordan & Dimakis, 2020). Instead, practical approaches rely on efficiently computable bounds or estimates as proxies (Latorre et al., 2020). Moreover, in a recent line of research, several studies have focused on establishing explicit upper and lower bounds for the Lipschitz constant of random neural networks, i.e., neural networks at initialization (Geuchen et al., 2025; Dirksen et al., 2025). The main limitation of these works is that they are restricted to characterizing the Lipschitz constant only at initialization, and do not account for how the Lipschitz constant evolves during training. Despite this growing body of theoretical work, accurately computing or tightly estimating the Lipschitz constant of overparameterized deep networks remains challenging.

### 1.1. Contribution

The objective of our work is to develop a deeper theoretical understanding of the Lipschitz continuity of modern overparameterized DNNs, with a particular focus on their behavior at initialization and around its vicinity. Specifically, we aim to derive explicit upper bounds on the Lipschitz constant of Fully Connected Networks (FCNs) that hold with high prob-

ability over the randomness of the initialization, without relying on any form of explicit regularization. Our primary contributions are summarized below:

1. By adopting a variant of the commonly used He initialization (He et al., 2015), we derive an upper bound on the Lipschitz constant of DNNs with smooth activations that does not depend on the width. This result implies that, although the network's expressive power increases with added neurons, its smoothness remains controlled.

2. Going beyond random initialization, we establish a uniform bound on the Lipschitz constant in the lazy training regime. This is done by relating perturbations of the network's Jacobian—measured in spectral norm—to perturbations of the weights. Leveraging this relationship, we show that the Lipschitz constant remains independent of the network's width, provided that the weights stay within a ball of radius $\mathcal{O}(R \cdot m^{-1/2})$ around their initialization, where $R > 0$ and $m$ denotes the network width. This result indicates that lazy training *implicitly* regularizes the network's Lipschitz constant.

3. To validate our theoretical bounds, we numerically estimate the Lipschitz constant at initialization and compare the results against both our predictions and existing certified Lipschitz estimation methods. We additionally consider a teacher–student setting that induces lazy training, demonstrating that the student network's Lipschitz constant remains stable as the network width increases. Finally, using the same setup, we study how network width affects adversarial robustness after training.

**Comparison to existing bounds:** A detailed comparison between our results and related prior work is presented in Table 1. The work of (Geuchen et al., 2025) establishes Lipschitz bounds for deep networks at initialization. Their upper bound grows exponentially with the network depth $L$, while exhibiting only a mild logarithmic dependence on the width $m$, namely $\mathcal{O}(\sqrt{\log(m)})$. More recently, (Dirksen et al., 2025) eliminated the exponential dependence on depth, although the logarithmic dependence on width remains. Both results are specific to networks with ReLU activations. In contrast, our results extend these guarantees to deep networks with smooth activations. While our bounds retain the exponential dependence on depth appearing in (Geuchen et al., 2025), they completely remove the logarithmic dependence on width. Moreover, our analysis naturally extends beyond initialization to networks trained in the lazy-training regime.

*Table 1.* Comparison of our results with existing Lipschitz bounds. Our estimates retain the exponential depth dependence in (Geuchen et al., 2025), remove the logarithmic dependence on the width, and extend beyond initialization to the lazy-training regime.

| Result | Activation | Initialization | Lazy Training |
|--------|------------|----------------|---------------|
| (Geuchen et al., 2025) | ReLU | $\mathcal{O}\left(2^{\mathcal{O}(L)}\sqrt{Ld\log(m/d)}\right)$ | – |
| (Dirksen et al., 2025) | ReLU | $\mathcal{O}\left(\sqrt{d\log(m/d)}\right)$ | – |
| **Ours** | Smooth | $\mathcal{O}\left(2^{\mathcal{O}(L)}\sqrt{d}\right)$ | $\mathcal{O}\left(2^{\mathcal{O}(L)}(\sqrt{d}+LR)\right)$ |

### 1.2. Organization

The remainder of the paper is organized as follows. In Section 2, we briefly review prior work, both theoretical and empirical in nature, related to the Lipschitz constant of neural networks. Section 3 introduces the specific DNN architecture considered, outlines the assumptions on the activation function and input space, and formally states the problem. The main theoretical results are presented in Section 4, followed by supporting numerical experiments in Section 6. Finally, Section 7 offers concluding remarks, discusses the potential impact of our findings, and outlines limitations and directions for future research.

## 2. Related Work

**Theoretical works:** In recent years, the study of Lipschitz continuity in neural networks has received increasing attention from the research community. This interest is driven in part by the seminal work of Bartlett et al. (Bartlett et al., 2017), which established a strong link between the Lipschitz constant and the generalization properties of DNNs via margin-based bounds, and by subsequent extensions of this framework (Chuang et al., 2021; Li et al., 2018). Another important motivation comes from the demonstrated vulnerability of DNNs to adversarial attacks (Bubeck et al., 2021; Wu et al., 2021; Carlini et al., 2019), where the Lipschitz constant plays a central role in certifying robustness. This has inspired numerous efforts to incorporate robustness into the training process, either through Lipschitz-aware regularization or by designing specialized architectures with controlled Lipschitz properties (Qian & Wegman, 2018; Tsuzuku et al., 2018). Despite its importance, it has been proven that, even for shallow neural networks consisting of a single hidden layer, exactly calculating the Lipschitz constant is NP-hard (Virmaux & Scaman, 2018; Jordan & Dimakis, 2020), motivating the development of efficient algorithmic approximation and bounding techniques (Ebihara et al., 2024; Fazlyab et al., 2019; Jordan & Dimakis, 2020; Latorre et al., 2020; Virmaux & Scaman, 2018). Notable examples of such techniques include *SeqLip* (Virmaux & Scaman, 2018), which estimates an upper bound on a network's Lipschitz constant by propagating layer-wise operator norm bounds through the network in a sequential

manner, and *LipSDP* (Fazlyab et al., 2019), which formulates the estimation as a semidefinite program (SDP) to provide tighter and more computationally feasible bounds. Another line of research (Buchanan et al., 2020; Dirksen et al., 2025; Geuchen et al., 2025) establishes probabilistic upper and lower bounds on the Lipschitz constant of random ReLU networks.

**Practical applications:** Numerous theoretical works are complemented by experimental studies that provide empirical insight, as well as ad-hoc algorithms designed to promote small Lipschitz constants and thereby improve the robustness of trained models. Interestingly, recent works (Gamba et al., 2023; Khromov & Singh, 2024) have observed a close correlation between the Lipschitz constant and the double-descent phenomenon in overparameterized systems. Specifically, the findings suggest that larger overparameterized models tend to implicitly regularize the Lipschitz constant. This observation aligns with the benign overfitting phenomenon—where models generalize well despite fitting the training data almost perfectly (Bartlett et al., 2020; Li et al., 2021; Tsigler & Bartlett, 2023). Other methods explicitly regularize the network's Lipschitz constant during training to improve robustness against adversarial perturbations and noisy inputs while maintaining high accuracy. For example, (Pauli et al., 2021) use semidefinite relaxations combined with ADMM optimization, whereas (Aziznejad et al., 2020) introduce trainable activation functions that enhance expressivity while keeping the Lipschitz constant under control. Finally, as alternative architectures such as Convolutional Neural Networks (CNNs) and Transformers built on self-attention have gained popularity, various approaches have been proposed to enforce Lipschitz regularity in these models as well (Li et al., 2019; Kim et al., 2021).

## 3. Preliminaries and Overview

### 3.1. Notation and Definitions

Bold lowercase letters denote vectors (e.g., $\mathbf{x}$), while uppercase letters denote matrices (e.g., $W$), or random variables (e.g., $X$), with the intended meaning inferred from context. The $(i, j)$-th element of a matrix is denoted by $[W]_{i,j}$. We use $[n]$ to denote the set $\{1, \ldots, n\}$. For the standard scalar product between two vectors $\mathbf{x}, \mathbf{y} \in \mathbb{R}^d$, we write $\langle \mathbf{x}, \mathbf{y} \rangle$, while the induced Euclidean norm is $\|\mathbf{x}\|_2$. The infinity norm of a vector is $\|\mathbf{x}\|_\infty$. For a matrix $W$, the Frobenius norm is denoted by $\|W\|_{\mathrm{F}}$, whereas the operator norm induced by $\|\cdot\|_\alpha$ and $\|\cdot\|_\beta$ is $\|W\|_{\alpha \to \beta}$. When $\alpha = \beta = 2$, the induced norm is the spectral norm, in which case, we drop the subscript notation and simply write $\|W\|$. The closed Euclidean ball centered at $\mathbf{x} \in \mathbb{R}^d$ with radius $R > 0$ is $\overline{\mathcal{B}}_d(\mathbf{x}, R)$. Given matrices $W_1, \ldots, W_\ell$, we write $\boldsymbol{\theta}_\ell := \mathrm{vec}(W_1, \ldots, W_\ell)$ to denote their vectorized form, obtained by stacking all matrices into a single vector, whereas the closed $R$ neighborhood centered around $\boldsymbol{\theta}_\ell$ is denoted by

$$\overline{\mathcal{B}}(\boldsymbol{\theta}_\ell, R) := \big\{ \mathrm{vec}(\widetilde{W}_1, \ldots, \widetilde{W}_\ell) \ :$$
$$\|\widetilde{W}_r - W_r\| \leq R, \forall r \in [\ell] \big\}.$$

### 3.2. Some Basic Facts about Lipschitz Functions

**Definition 3.1.** Let $\mathcal{X} \subset \mathbb{R}^{d_{\mathrm{in}}}$ and let $f : \mathbb{R}^{d_{\mathrm{in}}} \to \mathbb{R}^{d_{\mathrm{out}}}$ be a function. The local Lipschitz constant of $f$ over the set $\mathcal{X}$, with respect to the norms $\|\cdot\|_\alpha$ and $\|\cdot\|_\beta$, is defined as

$$\mathrm{Lip}_{(\alpha,\beta)}(f, \mathcal{X}) := \sup_{\substack{\mathbf{x}, \mathbf{y} \in \mathcal{X} \\ \mathbf{x} \neq \mathbf{y}}} \frac{\|f(\mathbf{x}) - f(\mathbf{y})\|_\beta}{\|\mathbf{x} - \mathbf{y}\|_\alpha}. \quad (1)$$

If $\mathrm{Lip}_{(\alpha,\beta)}(f, \mathcal{X}) < \infty$, we say that $f$ is locally Lipschitz continuous on the set $\mathcal{X}$.

It is easy to see that the Lipschitz constant measures the maximum rate at which the output of $f$ can change relative to changes in the input, across all pairs of distinct points $\mathbf{x}, \mathbf{y} \in \mathcal{X}$. Equivalently, $\mathrm{Lip}_{(\alpha,\beta)}(f, \mathcal{X})$ can be defined as the smallest positive constant $K$ such that the inequality $\|f(\mathbf{x}) - f(\mathbf{y})\|_\beta \leq K\|\mathbf{x} - \mathbf{y}\|_\alpha$ holds for all $\mathbf{x}, \mathbf{y} \in \mathcal{X}$. By setting $\mathcal{X} = \mathbb{R}^{d_{\mathrm{in}}}$, Equation (1) captures the *global* behavior of $f$. However, in this work we focus on the local Lipschitz constant, which provides a sharper and more informative measure than the often overly conservative global bound.

*Remark* 3.2. The above definition is consistent with the notion of a *local Lipschitz constant* commonly used in the machine learning literature (Jordan & Dimakis, 2020), where the term "local" refers to restricting the domain to an arbitrary subset $\mathcal{X} \subseteq \mathbb{R}^{d_{\mathrm{in}}}$. This differs from the classical notion of *local Lipschitz continuity* in analysis. A function $f$ is locally Lipschitz if, for every $\mathbf{x} \in \mathbb{R}^{d_{\mathrm{in}}}$, there exists a neighbourhood $U_\mathbf{x}$ of $\mathbf{x}$ and a constant $L_\mathbf{x} < \infty$ such that

$$\|f(\mathbf{u}) - f(\mathbf{v})\|_\beta \leq L_\mathbf{x}\|\mathbf{u} - \mathbf{v}\|_\alpha, \qquad \forall \mathbf{u}, \mathbf{v} \in U_\mathbf{x}.$$

Thus, the classical definition is pointwise and allows the Lipschitz constant to vary across neighborhoods, whereas Definition 3.1 associates a single constant to the entire set.

Even though Equation (1) provides an intuitive definition, it is generally difficult to compute in practice. However, it is well known that for a continuously differentiable function $f$, the Lipschitz constant is closely linked to the maximal operator norm of its Jacobian matrix. Specifically, for an open and convex set $\mathcal{X}$, this relationship is made explicit in (Dirksen et al., 2025). Often stated in the literature (somewhat loosely) as a consequence of Rademacher's theorem, it shows that

$$\mathrm{Lip}_{(\alpha,\beta)}(f, \mathcal{X}) = \sup_{\mathbf{x} \in \mathcal{X}} \|J_f(\mathbf{x})\|_{\alpha \to \beta}, \quad (2)$$

where $J_f(\mathbf{x}) \in \mathbb{R}^{d_{out} \times d_{in}}$ denotes the Jacobian of $f$ at $\mathbf{x}$. The expression above is much easier to handle, since it indicates that upper bounds for the Lipschitz constant can be established by bounding the operator norm of the Jacobian matrix. In case Euclidean norms are used in both input and output, the induced $\alpha$-to-$\beta$ Jacobian norm simply reduces to the spectral norm. Since this case will be our primary focus, we will, for the remainder of the paper, simplify the notation by dropping the subscript indices and write $\mathrm{Lip}(f, \mathcal{X})$.

*Remark* 3.3. We note that the convexity assumption on $\mathcal{X}$ is not strictly necessary for Equation (2) to provide an upper bound. Even for non-convex sets, the supremum of the Jacobian norm over $\mathcal{X}$ still serves as a valid (though potentially conservative) estimate of the Lipschitz constant. We will further elaborate on this fact in subsequent sections.

### 3.3. Network Architecture and Assumptions

The main objective of this work is to derive an upper bound on the Lipschitz constant, as defined in Equation (1), when the function $f$ corresponds to a fully-connected DNN. To this end, we first introduce the network architecture under consideration, define the relevant quantities, and specify the assumptions underlying our analysis. In particular, we detail the conditions imposed on the activation function, the input domain, and the random initialization scheme we adopt.

**Fully connected deep neural networks:** We consider a deep feed-forward neural network $f_{\boldsymbol{\theta}_{L+1}} : \mathbb{R}^{d_{in}} \to \mathbb{R}^{d_{out}}$, consisting of $L$ hidden layers that is recursively defined in the following way

$$
\begin{aligned}
f_{\boldsymbol{\theta}_{L+1}}(\mathbf{x}) &= \sqrt{m} \cdot W_{L+1}\, f_{\boldsymbol{\theta}_L}(\mathbf{x}), \\
f_{\boldsymbol{\theta}_\ell}(\mathbf{x}) &= \phi\left(W_\ell\, f_{\boldsymbol{\theta}_{\ell-1}}(\mathbf{x})\right), \quad \ell = 2, \ldots, L, \\
f_{\boldsymbol{\theta}_1}(\mathbf{x}) &= \phi\left(W_1 \mathbf{x}\right),
\end{aligned} \tag{3}
$$

where $\phi : \mathbb{R} \to \mathbb{R}$ is a nonlinear activation function applied component-wise and $\boldsymbol{\theta}_\ell := \mathrm{vec}(W_1, \ldots, W_\ell)$ for $\ell \in [L+1]$ are the vectorized representations of the network's parameters up to the $\ell$-th layer, with the first weight matrix $W_1 \in \mathbb{R}^{m \times d_{in}}$, the hidden layer matrices $W_\ell \in \mathbb{R}^{m \times m}$ for $\ell = 2, \ldots, L$, and the final weight matrix $W_{L+1} \in \mathbb{R}^{d_{out} \times m}$. For notational simplicity, we assume that the width of the network is the same across all hidden layers and is equal to $m$. The functions $f_{\boldsymbol{\theta}_\ell} : \mathbb{R}^{d_{in}} \to \mathbb{R}^m$ for $\ell \in [L]$ are called the *post-activation* mappings of the network and represent the outputs of the intermediate layers.

The properties of the network $f_{\boldsymbol{\theta}_{L+1}}$ heavily depend on the choice of activation function $\phi$. To derive our results, we impose the following conditions.

**Assumption 3.4.** (Activation function) The activation function $\phi : \mathbb{R} \to \mathbb{R}$ satisfies $\phi(0) = 0$, is 1-Lipschitz, and $\beta_\phi$-smooth; that is, there exists $\beta_\phi > 0$ such that for all $x, y \in \mathbb{R}$,

$$
\begin{aligned}
|\phi(x) - \phi(y)| &\leq |x - y|, \\
|\phi'(x) - \phi'(y)| &\leq \beta_\phi |x - y|.
\end{aligned} \tag{4}
$$

We note that the above assumption excludes the popular Rectified Linear Unit (ReLU) activation function, defined as $\phi(x) := \max\{0, x\}$, since it does not satisfy the required smoothness condition. This smoothness assumption, commonly adopted in many theoretical works (Jacot et al., 2018; Du et al., 2019), simplifies the analysis by ensuring well-behaved derivatives. Given that a large body of research has already extensively studied networks with ReLU activations (Geuchen et al., 2025; Buchanan et al., 2020), we do not address this case here.

**Examples:** Classical activation functions such as the hyperbolic tangent satisfy the assumptions outlined above. However, as a concrete example, we focus on the modern Sigmoid-Weighted Linear Unit (SiLU) activation function (Elfwing et al., 2018), which has recently gained popularity due to its strong empirical performance. It is explicitly defined as

$$
\phi(x; \beta_\phi) := \frac{\gamma^{-1} x}{1 + e^{-2\gamma\beta_\phi x}}, \tag{5}
$$

where $\gamma = \sup_{x \in \mathbb{R}}\left\{\frac{1}{1+e^{-x}} + \frac{xe^{-x}}{(1+e^{-x})^2}\right\} \approx 1.1$ is a scaling constant. Other popular examples of modern smooth activation functions include the Exponential Linear Units (ELU) (Clevert et al., 2015) and Gaussian Error linear Units (GeLU) (Hendrycks & Gimpel, 2016).

**Assumption 3.5.** (Random initialization) The weight matrices $W_1, \ldots, W_{L+1}$ are independently initialized using the following scheme:

$$
[W_\ell]_{i,j} \overset{\text{i.i.d.}}{\sim} \mathcal{N}\left(0, \tfrac{1}{m}\right), \quad \forall \ell \in [L+1]. \tag{6}
$$

The weights in the hidden layers are initialized as Gaussian random matrices with independent, zero-mean entries and variance inversely proportional to the network's width. Similar initialization strategies have been employed in several theoretical works (Du et al., 2019; Jacot et al., 2018; Lee et al., 2017), as it is well known that, in the limit as $m \to \infty$, the output of each layer converges to a GP (Lee et al., 2017).

**Assumption 3.6.** (Input set) We assume that the input set $\mathcal{X} \subset \mathbb{R}^{d_{in}}$ is bounded. That is, there exists some $\Lambda > 0$ such that $\|\mathbf{x}\|_2 \leq \Lambda$, for all $\mathbf{x} \in \mathcal{X}$.

The above boundedness assumption is trivially satisfied in most practical applications. This typically holds either because the underlying data distribution naturally has bounded support or because explicit normalization is applied as a standard preprocessing step to constrain the data within a fixed range.

### 3.4. Network Jacobian & the Naive Lipschitz Bound:

In view of the compositional structure of the network $f_{\boldsymbol{\theta}_{L+1}}$ of Equation (3), a simple application of the chain rule yields

$$J_{\boldsymbol{\theta}_{L+1}}(\mathbf{x}) = \sqrt{m} \cdot W_{L+1} \prod_{\ell=1}^{L} D_{\boldsymbol{\theta}_\ell}(\mathbf{x}) W_\ell, \qquad (7)$$

where $J_{\boldsymbol{\theta}_{L+1}}(\mathbf{x}) \in \mathbb{R}^{d_{\text{out}} \times d_{\text{in}}}$ is the network's Jacobian matrix, $\mathbf{x}_\ell := f_{\boldsymbol{\theta}_\ell}(\mathbf{x}) \in \mathbb{R}^m$ are the post activations, and $D_{\boldsymbol{\theta}_\ell}(\mathbf{x}) := \text{diag}(\phi'(W_\ell \mathbf{x}_{\ell-1})) \in \mathbb{R}^{m \times m}$ for $\ell = 1, \ldots, L$, where we make the convention $\mathbf{x}_0 := \mathbf{x}$. Furthermore, it is easy to see the Jacobian matrices of the post activation mappings are given by

$$J_{\boldsymbol{\theta}_\ell}(\mathbf{x}) = \prod_{r=1}^{\ell} D_{\boldsymbol{\theta}_r}(\mathbf{x}) W_r.$$

Equation (2) provides an exact characterization of the Lipschitz constant in terms of the Jacobian norm for open and convex sets. Nevertheless, even when these conditions are relaxed, it is possible to obtain meaningful upper bounds. Under Assumptions 3.4 and 3.6, by applying a straightforward sequence of inequalities and exploiting the properties of the spectral norm, we derive the following bound, whose proof can be found in Appendix B.1.

**Proposition 3.7.** *The local Lipschitz constant of the fully connected network $f_{\boldsymbol{\theta}_{L+1}}$, as defined in Equation (3), over the set $\mathcal{X}$ with respect to the Euclidean norm satisfies*

$$\text{Lip}(f_{\boldsymbol{\theta}_{L+1}}, \mathcal{X}) \leq \sup_{\mathbf{x} \in \overline{\mathcal{B}}_{d_{\text{in}}}(\mathbf{0}, \Lambda)} \|J_{\boldsymbol{\theta}_{L+1}}(\mathbf{x})\| \qquad (8a)$$

$$= \sup_{(\mathbf{x}, \mathbf{y}, \mathbf{z}) \in \mathcal{T}} \left\langle \mathbf{z}, \sqrt{m} \cdot W_{L+1} J_{\boldsymbol{\theta}_L}(\mathbf{x}) \mathbf{y} \right\rangle, \qquad (8b)$$

*where $\mathcal{T} := \overline{\mathcal{B}}_{d_{\text{in}}}(\mathbf{0}, \Lambda) \times \overline{\mathcal{B}}_{d_{\text{in}}}(\mathbf{0}, 1) \times \overline{\mathcal{B}}_{d_{\text{out}}}(\mathbf{0}, 1)$ and $J_{\boldsymbol{\theta}_L}(\mathbf{x}) \in \mathbb{R}^{m \times d_{\text{in}}}$ is the Jacobian matrix of $f_{\boldsymbol{\theta}_L}$.*

By substituting Equation (7) into (8a) an using the submultiplicativity of the spectral norm along with the 1-Lipschitz property of the activation function, we can immediately derive the following *global* bound

$$\text{Lip}(f_{\boldsymbol{\theta}_{L+1}}, \mathbb{R}^{d_{\text{in}}}) \leq \sqrt{m} \prod_{\ell=1}^{L+1} \|W_\ell\| =: K_{\text{naive}}(\boldsymbol{\theta}_{L+1}). \tag{9}$$

The following proposition quantifies the behavior of $K_{\text{naive}}(\boldsymbol{\theta}_{L+1})$ at initialization with respect to depth and width. The proof can be found in Appendix B.1.

**Proposition 3.8.** *Let $\boldsymbol{\theta}_{L+1} = \text{vec}(W_1, \ldots, W_{L+1})$ be a parameter vector initialized as in Assumption 3.5. If the network's width satisfies $m \geq \max\{d_{\text{in}}, d_{\text{out}}\}$, then*

$$K_{\text{naive}}(\boldsymbol{\theta}_{L+1}) \leq \sqrt{m} \cdot 3^{L+1},$$

*with probability at least $\left(1 - 2\exp(-cm)\right)^{L+1}$, where $c > 0$ is a universal constant.*

Despite its appealing simplicity, this estimate is relatively loose due to two primary shortcomings indicated by Proposition 3.8. First, it grows exponentially with the depth of the network. Second, at initialization it scales like $\mathcal{O}(\sqrt{m})$. Given that the most common and effective form of overparameterization involves increasing the network's width, this scaling fails to explain the remarkable success such wide networks have found in terms of generalization, whereas it contradicts recent experimental studies which observed that wider networks tend to have smaller Lipschitz constants (Khromov & Singh, 2024). Hence, we will refer to it as the *naive upper bound*. A central goal of this work is to eliminate the undesired $\sqrt{m}$ dependence.

### 3.5. Lazy Training & Lipschitz Perturbation

Theoretically analyzing the properties of neural networks at random initialization is the first step in understanding the principles that govern their behavior. However, as the distribution of the weights during training changes, it is necessary to extend the analysis beyond initialization to capture the network's evolving dynamics. To this end, and to facilitate the study of Lipschitz continuity throughout training, we introduce the following hypothesis class.

**Definition 3.9.** Let $W_1, \ldots, W_{L+1}$ be weight matrices initialized as in Assumption 3.5, and $R > 0$. Let $\boldsymbol{\theta}_{L+1} := \text{vec}(W_1, \ldots W_{L+1})$ be the corresponding parameter vector. We define the function class

$$\mathcal{F}(\boldsymbol{\theta}_{L+1}, R) := \Big\{ \mathbf{x} \mapsto f_{\tilde{\boldsymbol{\theta}}_{L+1}}(\mathbf{x}) :$$

$$\tilde{\boldsymbol{\theta}}_{L+1} \in \overline{\mathcal{B}}(\boldsymbol{\theta}_{L+1}, R \cdot m^{-1/2}) \Big\},$$

where $f_{\tilde{\boldsymbol{\theta}}_{L+1}}$ is the neural network defined as in Equation (3).

*Remark* 3.10. Even though $\mathcal{F}(\boldsymbol{\theta}_{L+1}, R)$ is reminiscent of the class of functions associated with the lazy training regime, we argue that it is, in fact, a broader class compared to the usual lazy training setup considered in other works. In particular, the authors in (Cao & Gu, 2019) introduce the Neural Tangent Random Feature (NTRF) class by considering all networks whose weights remain within a ball of radius $R \cdot m^{-1/2}$ centered at random initialization, measured in the Frobenius norm, and derive a width-independent generalization bound for this class. We note that the class $\mathcal{F}(\boldsymbol{\theta}_{L+1}, R)$ is a strict superclass of the NTRF, since the condition $\tilde{\boldsymbol{\theta}}_{L+1} \in \overline{\mathcal{B}}(\boldsymbol{\theta}_{L+1}, R \cdot m^{-1/2})$ only implies that $\|\widetilde{W}_\ell - W_\ell\|_F \leq R$ for all $\ell \in [L+1]$. That is, in our setting, we do not require that the Frobenius distance of the weights from their initialization shrinks with increasing width. Instead, we only require that this distance remains bounded.

# 4. Main results

This section presents our main theoretical contributions which can be traced in two theorems. Theorem 4.1, establishes a bound on the Lipschitz constant of a DNN at random initialization that holds with high probability, whereas Theorem 4.2 uniformly bounds the Lipschitz constant of the function class $\mathcal{F}(\boldsymbol{\theta}_{L+1}, R)$. The proofs for the two statements can be found in Appendices B.3 and B.5, respectively.

## 4.1. Lipschitz Constant Bound at Initialization

**Theorem 4.1.** *Let $\boldsymbol{\theta}_{L+1} = \mathrm{vec}(W_1, \ldots, W_{L+1})$ be parameter vector, and let $f_{\boldsymbol{\theta}_{L+1}} : \mathbb{R}^{d_{\mathrm{in}}} \to \mathbb{R}^{d_{\mathrm{out}}}$ be the corresponding network defined in Equation (3). Furthermore, let Assumptions 3.4, 3.5, and 3.6 hold. Then, for any $u \geq 0$, if the network's width satisfies $m = \Omega\big(\max\{d_{\mathrm{in}}, L^2 \log m\}\big)$, the local Lipschitz constant of $f_{\boldsymbol{\theta}_{L+1}}$ over the set $\mathcal{X}$ with respect to the Euclidean metric satisfies*

$$
\mathrm{Lip}(f_{\boldsymbol{\theta}_{L+1}}, \mathcal{X}) \leq C \cdot 2^L \Big[ \big(e\, 2^L \beta_\phi \Lambda + 1\big)
$$
$$
\times (\sqrt{d_{\mathrm{in}}} + u) + \sqrt{d_{\mathrm{out}}} \Big], \tag{10}
$$

*with probability at least $\big(1 - 2\exp(-u^2)\big)\big(1 - 4\exp(-cm/L^2)\big)^L$ over the random initialization, where $C, c > 0$ are universal constants.*

The first thing we note is that the Lipschitz constant at initialization behaves like $\mathcal{O}(\sqrt{d_{\mathrm{in}}} + \sqrt{d_{\mathrm{out}}})$, which is reminiscent to the spectral norm of a random Gaussian matrix of size $d_{\mathrm{out}} \times d_{\mathrm{in}}$ with independent entries (Vershynin, 2018). Moreover, the failure probability of this event decays as $m \to \infty$, which, in our view, provides further justification that infinitely wide neural networks behave like Gaussian processes at initialization. Most importantly, the upper bound of Equation (10) does *not* depend on $m$, suggesting that the continuity of DNNs remains unaffected by their width. Similar results exist for ReLU networks (Buchanan et al., 2020), however, to the best of our knowledge, we are the first to provide a proof for this fact for a general class of smooth activations.

Other works (Geuchen et al., 2025; Dirksen et al., 2025) have complemented upper bounds with corresponding lower bounds on the Lipschitz constant of ReLU networks at initialization. However, under the assumptions we impose, such lower bounds cannot be established. The reason is that our setting allows for trivial activation functions, such as the function that is identically zero for all $x \in \mathbb{R}$, which is both Lipschitz continuous and smooth. When used as the activation in a neural network, the entire network computes the zero function, and its corresponding Lipschitz constant is equal to zero.

## 4.2. Lipschitz Constant Bound During Lazy Training

**Theorem 4.2.** *Let $\boldsymbol{\theta}_{L+1} = \mathrm{vec}(W_1, \ldots, W_{L+1})$ be a parameter vector and suppose that Assumptions 3.4, 3.5, and 3.6 hold. If the network width satisfies $m = \Omega\big(\max\{d_{\mathrm{in}}, d_{\mathrm{out}}, L^2 \log m\}\big)$, then it holds that for any $u \geq 0$,*

$$
\sup_{h \in \mathcal{F}(\boldsymbol{\theta}_{L+1}, R)} \mathrm{Lip}(h, \mathcal{X}) \leq K_{\mathrm{init}} + C_{\mathrm{per}} \cdot R, \tag{11}
$$

*with probability at least $\big(1 - 2\exp(-u^2) - 2\exp(-cm)\big)\big(1 - 4\exp(-cm/L^2)\big)^L$. Here,*

$$
K_{\mathrm{init}} := C \cdot 2^L \Big[ \big(e\, 2^L \beta_\phi \Lambda + 1\big)
$$
$$
\times (\sqrt{d_{\mathrm{in}}} + u) + \sqrt{d_{\mathrm{out}}} \Big], \tag{12}
$$

*is the Lipschitz bound due to the effect of the random initialization and*

$$
C_{\mathrm{per}} := 3L \cdot 2^L \bigg[ 2^{L+1} \beta_\phi \Lambda \Big(\frac{R}{\sqrt{m}}\Big)^{2L} \max\Big\{1, e^2 \cdot \Big(\frac{2\sqrt{m}}{R}\Big)^{2L}\Big\}
$$
$$
+ \Big(\frac{R}{\sqrt{m}}\Big)^{L} \max\Big\{1, e \cdot \Big(\frac{2\sqrt{m}}{R}\Big)^{L}\Big\} \bigg]
$$
$$
+ 2^L \Big(\frac{R}{\sqrt{m}}\Big)^{L} \max\Big\{1, e \cdot \Big(\frac{2\sqrt{m}}{R}\Big)^{L}\Big\}, \tag{13}
$$

*controls the effect of the weight perturbations, where $C, c > 0$ are universal constants.*

Theorem 4.2 provides a uniform bound on the Lipschitz constant of the function class $\mathcal{F}(\boldsymbol{\theta}_{L+1}, R)$. The bound comprises two distinct terms. The first term, $K_{\mathrm{init}}$, corresponds to the Lipschitz bound at initialization, and is identical to the one appearing in Equation (10). The second term captures the effect of weight perturbations. In particular, if the width of the network is sufficiently large, that is, when $m = \Omega(R^2)$, we have $C_{\mathrm{per}} = \mathcal{O}(1)$. This means that it will remain constant as the width $m$ increases. Consequently, the overall Lipschitz constant of the network scales as $\mathcal{O}(R)$ and is unaffected by the network's width.

*Remark* 4.3. Although Theorem 4.1 could be stated as a corollary of Theorem 4.2 (by taking the limit $R \to 0$), we present them separately for clarity of exposition and because the bound in Theorem 4.2 requires additional restrictions to hold.

# 5. Key Ideas and Proof Ingredients

This section provides an overview of the proofs of Theorems 4.1 and 4.2. To derive the Lipschitz bound at initialization, we adopt the proof technique introduced in (Geuchen et al., 2025) for random ReLU networks, whereas for the

bound obtained during lazy training, we make use of a set of perturbation inequalities for networks with smooth activations, established in Appendix C.

## 5.1. Lipschitz Bound as the Supremum of a GP

By conditioning on the hidden layer weight matrices $W_1, \ldots, W_L$, the quantity inside the supremum of Equation (8b) becomes a Gaussian stochastic process indexed by elements of the high-dimensional set $\mathcal{T}$. This is made clear since the joint distribution of any finite collection $\{\langle \mathbf{z}_i, \sqrt{m} \cdot W_{L+1} J_{\boldsymbol{\theta}_L}(\mathbf{x}_i)\mathbf{y}_i\rangle\}_{=1}^{N}$ is a multi-variate Gaussian for any $\{(\mathbf{x}_i, \mathbf{y}_i, \mathbf{z}_i)\}_{i=1}^{N} \subset \mathcal{T}$. Hence, an upper bound for the network's local Lipschitz constant can be obtained by bounding the supremum of this Gaussian process. It is well known that the supremum of a stochastic process is closely tied to its continuity properties and the complexity of its index set. Leveraging this connection, we employ a *chaining* argument to derive a Lipschitz bound at initialization. The necessary mathematical background is presented in Appendix A.

Our proof can be split in two steps. First, after verifying that the necessary conditions are satisfied, we apply Dudley's integral (Vershynin, 2018) to show that for any $u \geq 0$, the inequality

$$\mathrm{Lip}(f, \mathcal{X}) \leq C \left[ \int_0^\infty \sqrt{\log N(\mathcal{T}, \rho, \varepsilon)} \, d\varepsilon + u \cdot \mathrm{diam}(\mathcal{T}) \right],$$

holds with high probability over the randomness of $W_{L+1}$, where $N(\mathcal{T}, \rho, \varepsilon)$ is the covering number of the set $\mathcal{T}$, $\rho : \mathcal{T} \times \mathcal{T} \to \mathbb{R}_{\geq 0}$ is an appropriate metric defined in Lemma B.2, and $C > 0$ is a constant. Next, we establish a bound for the covering number $N(\mathcal{T}, \rho, \varepsilon)$ that depends on the norms $\|W_\ell\|, \|W_\ell\|_{2\to\infty}$ for all $\ell \in [L]$, but not on the width $m$. Finally, to reintroduce the randomness of the remaining weight matrices, we bound the probabilities of the events

$$\mathcal{W}_{\mathrm{op}} := \left\{ \|W_\ell\| \leq 2 + \frac{2}{L}, \ \forall \ell \in [L] \right\},$$

$$\mathcal{W}_{2\to\infty} := \left\{ \|W_\ell\|_{2\to\infty} \leq 1 + \frac{1}{L}, \ \forall \ell \in [L] \right\},$$

and apply Proposition C.1 from (Geuchen et al., 2025) to combine the two bounds and obtain an explicit overall probabilistic bound for the Lipschitz constant.

## 5.2. Controlling the Lipschitz Perturbations

Having established a width-independent bound at initialization, we next extend this result to the lazy training regime of Definition 3.9. First, we show in Lemma B.6 that for any

two sets of parameters $\boldsymbol{\theta}_{L+1}$ and $\tilde{\boldsymbol{\theta}}_{L+1}$, it holds that

$$\mathrm{Lip}(f_{\tilde{\boldsymbol{\theta}}_{L+1}}, \mathcal{X}) \leq \sup_{(\mathbf{x}, \mathbf{y}, \mathbf{z}) \in \mathcal{T}} \left\langle \mathbf{z}, \sqrt{m} \cdot W_{L+1} J_{\boldsymbol{\theta}_L}(\mathbf{x})\mathbf{y} \right\rangle$$
$$+ \kappa\left(\boldsymbol{\theta}_{L+1}, \tilde{\boldsymbol{\theta}}_{L+1}\right),$$

where $\kappa\left(\boldsymbol{\theta}_{L+1}, \tilde{\boldsymbol{\theta}}_{L+1}\right)$ is a term that represents the effect of weight perturbations. Since by Theorem 4.1 we already know that the first term does not depend on the width, we focus on establishing a width-independent bound for the perturbation term. We follow a similar approach—as before—and show in Lemmas B.7 and B.8 that wherever $\boldsymbol{\theta}_L \in \mathcal{W}_{\mathrm{op}} \cap \mathcal{W}_{2\to\infty}$, and the spectral norm of the final layer satisfies $\|W_{L+1}\| \leq 3$, we have

$$\sup_{\tilde{\boldsymbol{\theta}}_{L+1} \in \overline{\mathcal{B}}(\boldsymbol{\theta}_{L+1}, \frac{R}{\sqrt{m}})} \kappa\left(\boldsymbol{\theta}_{L+1}, \tilde{\boldsymbol{\theta}}_{L+1}\right) \leq C_{\mathrm{per}} \cdot R,$$

where $C_{\mathrm{per}}$ is defined as in Theorem 4.2. Finally, to estimate the probability of the overall bound, we once more apply Proposition C.1 of (Geuchen et al., 2025).

# 6. Numerical Experiments

In this section, we investigate the behavior of the network's Lipschitz constant through a series of numerical experiments. We first validate Theorem 4.1 by examining how the Lipschitz constant at random initialization scales with key architectural and functional parameters—specifically, activation smoothness and input domain size—across networks of varying widths. We then compare these empirical observations with our theoretical bounds. Finally, we validate Theorem 4.2 using a teacher-student training setup, demonstrating that when the network operates in the lazy training regime, the Lipschitz constant remains well-controlled.

## 6.1. Empirical Investigation at Initialization

To analyze the Lipschitz constant at initialization, we fix the input dimension to $d_{\mathrm{in}} = 2$ and construct a fine uniform grid $\mathcal{G}_\delta$ with resolution $\delta > 0$ over the two-dimensional cube $\mathcal{X} := [-\Lambda, \Lambda]^2$. Then, we obtain an estimate for the Lipschitz constant through

$$K_{\mathrm{emp}}(\boldsymbol{\theta}_{L+1}) := \max_{\substack{\mathbf{x}, \mathbf{y} \in \mathcal{G}_\delta \\ \mathbf{x} \neq \mathbf{y}}} \frac{\left| f_{\boldsymbol{\theta}_{L+1}}(\mathbf{x}) - f_{\boldsymbol{\theta}_{L+1}}(\mathbf{y}) \right|}{\|\mathbf{x} - \mathbf{y}\|_2}, \quad (14)$$

where $f_{\boldsymbol{\theta}_{L+1}} : \mathbb{R}^2 \to \mathbb{R}$ is the network defined in Equation (3) and initialized according to Assumption 3.5. By construction, $K_{\mathrm{emp}}$ provides a *lower bound* for the true Lipschitz constant, since $\mathcal{G}_\delta \subset \mathcal{X}$. Nevertheless, previous studies (Khromov & Singh, 2024) suggest that the actual Lipschitz constant of neural networks is often much closer to this empirical lower bound than to the naive upper bound given in Equation (9).

**Experiment 1.** First, we evaluate the dependence of the empirical Lipschitz constant on the setup's hyperparameters. The results are presented in Figure 1. For each case, we evaluate the estimate in (14) using network widths $m \in \{32, 64, 128, 256\}$. Each reported value corresponds to the mean computed over 100 independent realizations of the network's initialization.

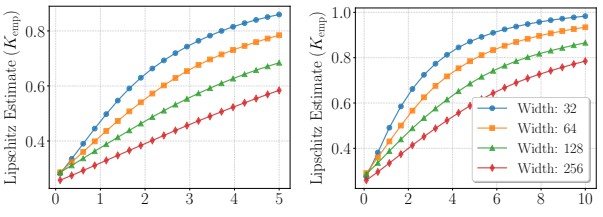

(a) Effect of smoothness ($\beta_\phi$)  (b) Effect of domain size ($\Lambda$)

*Figure 1.* Comparison of Lipschitz estimates under variations of key parameters: a) activation smoothness and b) domain size.

*Smoothness:* To investigate the effect of activation smoothness, we uniformly sample 20 values of the smoothness parameter $\beta_\phi$ ranging from 0.1 to 5, and construct the corresponding networks using the activation function defined in Equation (5). We set $\Lambda = 1.0$, the grid resolution to $\delta = 0.05$, and the number of hidden layers to $L = 2$. We observe that the empirical estimate increases with larger values of $\beta_\phi$, which is in agreement with the theoretical bound provided in Theorem 4.1.

*Domain size:* Similarly, for the domain size $\Lambda$, we uniformly sample 20 values in the range $[0.1, 10]$. For each value of $\Lambda$, the grid resolution is set to $\delta = \Lambda/10$, while the activation smoothness is fixed at $\beta_\phi = 1.0$. The number of hidden layers is set to $L = 2$. As shown in Figure 1(b), the Lipschitz estimate exhibits an increasing trend with respect to the domain size, consistent across different network widths.

**Obesrvations:** Overall, we find that the scaling of the empirical estimate of Equation (14) aligns with the theoretical predictions of Theorem 4.1 in both cases. Concerning the dependence on the width $m$, contrary to the naive bound in Equation (9), the empirical Lipschitz constant assumes *smaller* values as the width increases, rather than larger ones.

**Experiment 2.** Next, we compare the behavior of three distinct Lipschitz constant estimates, namely, the *naive* estimate, the *empirical* estimate, and the *EcLipSE* method (Xu & Sivaranjani, 2024), which formulates the Lipschitz constant estimation as a Semi Definite Program (SDP). We fix the network's depth to 3 and compare the different estimators for various widths. The results are reported in Table 1.

*Table 2.* Comparison of the *naive* Lipschitz estimate, the certified *EcLipSE* method, and the *empirical* estimate across different network widths.

| **Width:** $m$ | **32** | **64** | **128** | **256** | **512** | **1024** |
|---|---|---|---|---|---|---|
| $K_{\text{naive}}(\boldsymbol{\theta}_3)$ | 43.63 | 89.95 | 178.24 | 348.92 | 720.92 | 1456.80 |
| $K_{\text{EcLipSE}}(\boldsymbol{\theta}_3)$ | 8.00 | 12.24 | 17.58 | 23.31 | 34.19 | 49.43 |
| $K_{\text{emp}}(\boldsymbol{\theta}_3)$ | 0.47 | 0.39 | 0.29 | 0.31 | 0.32 | 0.27 |

**Observations:** The naive bound significantly overestimates the Lipschitz constant and exhibits the expected growth with increasing network width. The EcLipSE method provides a much tighter upper bound whose dependence on width is considerably weaker, though still increasing. In contrast, the empirical estimates are substantially smaller and decrease as the network width increases, in agreement with the predicted width-independence established in Theorem 4.1.

### 6.2. Empirical Investigation During Lazy Training

To study how the Lipschitz constant evolves during training, we consider a *teacher–student* setup in which the student network is trained to reproduce the outputs of the teacher. Specifically, let $\boldsymbol{\theta}_s = \text{vec}(W_3^s, W_2^s, W_1^s)$ and $\boldsymbol{\theta}_t = \text{vec}(W_3^t, W_2^t, W_1^t)$ denote the parameters of the teacher and student network, respectively, which are initialized according to Assumption 3.5. We define the corresponding two hidden layer networks

$$f_{\boldsymbol{\theta}_s}(\mathbf{x}) := \sqrt{m_s} \cdot W_3^s \phi(W_2^s \phi(W_1^s x)),$$

and

$$f_{\boldsymbol{\theta}_t}(\mathbf{x}) := \sqrt{m_t} \cdot W_3^t \phi(W_2^t \phi(W_1^t x)),$$

where $m_s$ and $m_t$ are the widths, with $W_3^s \in \mathbb{R}^{1 \times m_s}$, $W_2^s \in \mathbb{R}^{m_s \times m_s}$, $W_1^s \in \mathbb{R}^{m_s \times 1}$, and analogously $W_3^t \in \mathbb{R}^{1 \times m_t}$, $W_2^t \in \mathbb{R}^{m_t \times m_t}$, $W_1^t \in \mathbb{R}^{m_t \times 1}$. The activation $\phi(\cdot)$ is the SiLU of Equation (5) with $\beta_\phi = 1$, whereas we set $\Lambda = 1$.

**Experiment 3.** We generate $n = 128$ training samples according to the uniform distribution $x_1, \ldots, x_n \overset{\text{i.i.d.}}{\sim} \mathcal{U}([-50, 50])$ and corresponding labels $y_i = f_{\boldsymbol{\theta}_t}(x_i) + \sigma \cdot \zeta_i$ for $i \in [n]$, where $\sigma = 0.5$ and $\zeta_1, \ldots, \zeta_n \overset{\text{i.i.d.}}{\sim} \mathcal{N}(0, 1)$ are white additive noise variables. We keep the parameters of the teacher network fixed, and train the student network with full-batch gradient descent

$$\boldsymbol{\theta}_s(k+1) = \boldsymbol{\theta}_s(k) - \eta \nabla \mathcal{L}(\boldsymbol{\theta}_s(k)), \quad \forall k \in [T],$$

to minimize the mean squared error loss

$$\mathcal{L}(\boldsymbol{\theta}_s(k)) := \frac{1}{n} \sum_{i=1}^{n} (f_{\boldsymbol{\theta}_s(k)}(x_i) - y_i)^2,$$

where $\eta > 0$ is the learning rate and $\boldsymbol{\theta}_s(0)$ are the student network parameters at initialization. Af-

*Table 3.* Empirical Lipschitz constants, adversarial losses, and training losses for networks of different widths.

| Width: $m_s$ | 32 | 64 | 128 | 256 | 512 | 1024 |
|---|---|---|---|---|---|---|
| $\mathcal{L}(\boldsymbol{\theta}_s(T))$ | 0.23 | 0.22 | 0.22 | 0.22 | 0.22 | 0.23 |
| $\mathcal{L}^\epsilon_{\text{adv}}(\boldsymbol{\theta}_s(T))$ | 13.04 | 13.00 | 13.01 | 12.98 | 12.99 | 13.04 |
| $K_{\text{emp}}(\boldsymbol{\theta}_s(T))$ | 1.25 | 1.21 | 1.25 | 1.23 | 1.22 | 1.22 |

ter training for $T = 5 \times 10^4$ iterations with learning rate $\eta = 2 \times 10^{-4}$, we measure the deviation of the learned weights from initialization using the relative norms, $r_{\text{F}}(T, \ell) := \|W^s_\ell(T) - W^s_\ell(0)\|_{\text{F}} / \|W^s_\ell(0)\|_{\text{F}}$ and $r_{\text{op}}(T, \ell) := \|W^s_\ell(T) - W^s_\ell(0)\| / \|W^s_\ell(0)\|$, respectively, for $\ell = 1, 2, 3$. We evaluate these quantities for $m_s \in \{32, 64, 128, 256, 512, 1024\}$ with $m_t = 256$, and estimate the local Lipschitz constant via Equation (14). Figure 2 and Table 3 report the results, showing averages computed over 20 independent random initializations.

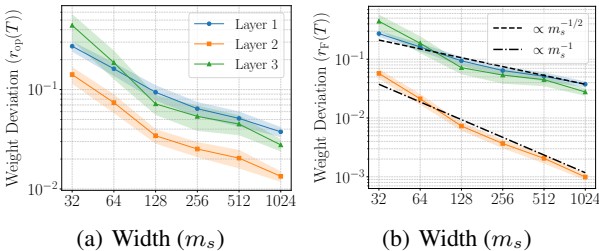

(a) Width ($m_s$)   (b) Width ($m_s$)

*Figure 2.* Relative per-layer weight deviations from the initialization: a) $r_{\text{op}}(T, \ell)$ and b) $r_{\text{F}}(T, \ell)$, for $\ell = 1, 2, 3$ as measured with respect to the spectral and Frobenius norms.

Additionally, we investigate the effect of width on adversarial robustness. After training, we evaluate the adversarial robustness of the learned student models by computing the average adversarial loss

$$\mathcal{L}^\epsilon_{\text{adv}}(\boldsymbol{\theta}_s) := \frac{1}{N} \sum_{i=1}^N \max_{\hat{x} \in \mathcal{S}_i(\epsilon)} \ell(f_{\boldsymbol{\theta}_s}(\hat{x}), f_{\boldsymbol{\theta}_t}(x_i)),$$

where $\ell(\cdot, \cdot)$ denotes the squared loss and $\mathcal{S}_i(\epsilon) := \overline{\mathcal{B}}(x_i, \epsilon)$ is the $\ell_\infty$-ball of radius $\epsilon$ centered at the input $x_i$. We sample $N = 512$ test inputs independently from the same distribution used during training, and generate adversarial perturbations using Projected Gradient Descent (PGD) (Madry et al., 2017) with perturbation budget $\epsilon = 5$, step size $\alpha = 0.25$, and 40 attack iterations. Since the choice of teacher initialization can significantly influence the resulting robustness independently of the student width, we repeat the experiment over five independently generated teacher networks and report averaged results. The results are reported in Table 3.

**Observations:** We observe that $r_{\text{op}}(T, \ell) = \mathcal{O}(m_s^{-1/2})$ for all $\ell$, providing numerical evidence that the student network

operates in the lazy regime throughout training. Consequently, the learned model lies in the class $\mathcal{F}(\boldsymbol{\theta}_s(0), R)$ for some $R > 0$. Moreover, Table 3 shows that the Lipschitz constant does not increase with network width, consistent with the prediction of Theorem 4.2. In all cases, the training loss falls slightly below the noise level $\sigma^2 = 0.25$. We further observe in Figure 2b that $r_{\text{F}}(T, 2) = \mathcal{O}(m_s^{-1})$, implying that the student network remains in the NTRF function class during training (Cao & Gu, 2019). Since this regime is a strict subclass of $\mathcal{F}(\boldsymbol{\theta}_s(0), R)$, our width-independent Lipschitz bounds apply *a fortiori*, indicating that our assumptions are conservative relative to what is observed in practice. Regarding the robustness, we observe that the adversarial loss remains stable across different network widths, suggesting that overparameterization does not degrade adversarial robustness in the lazy training regime. This behavior is consistent with the width-independent Lipschitz bounds established in Theorem 4.2.

# 7. Concluding Remarks

We derived explicit, width-independent upper bounds on the local Lipschitz constant of fully connected networks, both at initialization and in the lazy-training regime. Our analysis shows that sufficiently wide networks are inherently smooth, with Lipschitz constants comparable to those of Gaussian matrices, and that this smoothness persists as long as the parameters remain close to their initialization; these findings are supported by numerical experiments. A key limitation of our work is that we do not optimize the dependence of the bounds on network depth, which remains exponential, as is typical in Lipschitz-based analyses of deep architectures. Improving this depth dependence—potentially by exploiting additional structure of specific activation functions—remains an open direction. Another limitation is our reliance on smooth activations. While this assumption is essential for controlling Jacobian perturbations in the lazy-training analysis, we conjecture that analogous width-independent behavior should also hold for ReLU networks. Future work includes extending these results to other architectures, and more general training dynamics.

## Impact Statement

This work advances the theoretical understanding of stability in wide deep networks by clarifying how their Lipschitz constants behaves at and near initialization. These insights may inform the design of models with improved robustness and generalization. As a purely theoretical study, the direct societal impact is limited, though applications of Lipschitz-based robustness analyses in safety-critical settings may warrant careful consideration.

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

# A. Subgaussian Processes and Dudley's Inequality

This section presents the mathematical background and key definitions required to state Dudley's integral inequality, which serves as the central mathematical tool used for our proofs throughout this paper. We start by formalizing the notion of a subgaussian random variable and its subgaussian norm with the following definition.

**Definition A.1.** (Definition 2.6.4 in (Vershynin, 2018)) A random variable $X$ is called subgaussian if it possesses a finite subgaussian norm, defined by

$$\|X\|_{\psi_2} := \inf\{K > 0 \ : \ \mathbb{E}\exp(X^2/K^2) \le 2\}.$$

Consequently, its tail probabilities satisfy

$$\mathbb{P}(|X| > u) \le 2\exp(-c\,u^2/\|X\|_{\psi_2}^2) \quad \text{for all } u \ge 0,$$

for some absolute constant $c > 0$.

As Definition A.1 indicates, a subgaussian random variable is characterized by tail probabilities that decay at least as rapidly as those of a Gaussian. This class encompasses many familiar examples, including bounded random variables, Gaussian variables, and symmetric Bernoulli (Rademacher) variables. Their importance stems from the strong *concentration* properties they exhibit. The following proposition, which serves as the analogue of the Pythagorean theorem for the subgaussian norm, plays a central role in our proofs and is therefore stated here without proof for the sake of completeness.

**Proposition A.2.** *(Proposition 2.7.1 in (Vershynin, 2018)) Let $X_1, \ldots, X_N$ be independent, mean-zero, subgaussian random variables. Then,*

$$\left\|\sum_{i=1}^{N} X_i\right\|_{\psi_2}^2 \le C \sum_{i=1}^{N} \|X_i\|_{\psi_2}^2,$$

*where $C > 0$ is an absolute constant.*

A stochastic process $(X_t)_{t \in \mathcal{T}}$ is a family of random variables defined on a common probability space and indexed by the elements of a set $\mathcal{T}$. One way to study the continuity of such a process is through the subgaussian norm of its increments.

**Definition A.3.** (Definition 8.1.1 in (Vershynin, 2018)) Let $(X_t)_{t \in \mathcal{T}}$ be a stochastic process on a metric space $(\mathcal{T}, \rho)$. We say that the process has subgaussian increments (or is $K_S$-subgaussian) if there exists $K_S > 0$ such that

$$\|X_t - X_s\|_{\psi_2} \le K_S \rho(t, s) \quad \text{for all } t, s \in \mathcal{T}.$$

Equivalently, it holds that for all $u \ge 0$

$$\mathbb{P}(|X_t - X_s| > u) \le 2\exp\left(-\frac{cu^2}{K_s^2 \rho(t,s)^2}\right) \quad \text{for all } t, s \in \mathcal{T},$$

for some absolute constant $c > 0$.

Proposition 3.7 associates the network's Lipschitz constant to the supremum of a Gaussian process. In analyzing such suprema, two fundamental concepts—*covering numbers* and *metric entropy*—play a central role and are thus formalized in the following definition.

**Definition A.4.** (Definition 4.2.1 (Vershynin, 2018)) Let $(\mathcal{T}, \rho)$ be a metric space. Consider a subset $\mathcal{S} \subset \mathcal{T}$ and let $\varepsilon > 0$. A subset $\mathcal{G} \subseteq \mathcal{S}$ is called an $\varepsilon$-net or $\varepsilon$-cover of $\mathcal{S}$ if

$$\forall x \in \mathcal{S}, \ \exists\, y \in \mathcal{G} \ : \ \rho(x, y) \le \varepsilon.$$

The smallest possible cardinality of an $\varepsilon$-net of $\mathcal{S}$ is called the covering number, denoted by $N(\mathcal{S}, \rho, \varepsilon)$. The logarithm of the covering number, $\log N(\mathcal{S}, \rho, \varepsilon)$, is referred to as the metric entropy of $\mathcal{S}$.

From an information-theoretic perspective, metric entropy quantifies the number of bits required to specify any element of a set to within a given accuracy $\varepsilon$. In this sense, it serves as a measure of the set's complexity, reflecting how much information is needed to describe its elements at a prescribed level of precision.

Bounding the covering number of an arbitrary set can be a challenging task; however, for subsets of the unit ball, the covering number has been well studied, and explicit upper bounds based on a volumetric argument can be established, as formalized in the following proposition.

**Proposition A.5.** *(Proposition C.3 in (Foucart & Rauhut, 2013)) Let $\|\cdot\|$ be some norm on $\mathbb{R}^d$ and let $\mathcal{S}$ be a subset of the unit ball $\overline{\mathcal{B}}_d(\mathbf{0}, 1)$. Then, the covering number satisfies, for $\varepsilon > 0$*

$$N(\mathcal{S}, \|\cdot\|, \varepsilon) \leq \left(1 + \frac{2}{\varepsilon}\right)^d.$$

Having introduced the necessary definitions and preliminary results, we are now in a position to state Dudley's integral inequality. This inequality provides a powerful upper bound on the expected supremum of a subgaussian process in terms of the metric entropy of the underlying index set.

**Theorem A.6.** *(Theorem 8.1.3 in (Vershynin, 2018)) Let $(X_t)_{t \in \mathcal{T}}$ be a mean-zero stochastic process indexed by the metric space $(\mathcal{T}, \rho)$. If the process is $K_S$-subgaussian, then for any $u \geq 0$, the event*

$$\sup_{t,s \in \mathcal{T}} |X_t - X_s| \leq C K_S \left[\int_0^\infty \sqrt{\log N(\mathcal{T}, \rho, \varepsilon)}\, d\varepsilon + u \cdot \mathrm{diam}(\mathcal{T})\right], \tag{15}$$

*holds with probability at least $1 - 2\exp(-u^2)$, where $\mathrm{diam}(\mathcal{T}) := \sup_{t,s \in \mathcal{T}} \rho(t, s)$ denotes the diameter of $\mathcal{T}$, and $C > 0$ is an absolute constant.*

# B. Establishing Upper Bounds for the Lipschitz Constant

With the main mathematical tools established, we now proceed to the detailed proofs of the paper's central results. This section is organized around two primary objectives: (i) deriving bounds on the Lipschitz constant at initialization (Theorem 4.1), and (ii) analyzing its behavior in a neighborhood around the initialization (Theorem 4.2).

For the remainder of this paper, we will use the following notation: Given two sets of weight matrices $W_1, \ldots, W_{L+1}$ and $\widetilde{W}_1, \ldots, \widetilde{W}_{L+1}$ with compatible dimensions, we denote the corresponding parameter vectors up to the network's $\ell$-th layer as $\boldsymbol{\theta}_\ell = \mathrm{vec}(W_1, \ldots, W_\ell)$, and $\tilde{\boldsymbol{\theta}}_\ell = \mathrm{vec}(\widetilde{W}_1, \ldots, \tilde{W}_\ell)$ respectively, for all $\ell \in [L+1]$. Moreover, for a given input $\mathbf{x} \in \mathcal{X}$, we will use $\mathbf{x}_\ell = f_{\boldsymbol{\theta}_\ell}(\mathbf{x})$ and $\tilde{\mathbf{x}}_\ell = f_{\tilde{\boldsymbol{\theta}}_\ell}(\mathbf{x})$ to denote the values of the post-activations.

Since the connection between the Lipschitz constant and the supremum of a Gaussian process, established in Proposition 3.7, constitutes the key insight underlying our subsequent derivations, we begin by presenting a proof of this result.

## B.1. Proof of Proposition 3.7

By assumption, we know that the input set is bounded, that is, $\mathcal{X} \subset \overline{\mathcal{B}}_{d_{\mathrm{in}}}(\mathbf{0}, \Lambda)$. Hence, by the definition of the local Lipschitz constant, we can derive the following upper bound

$$\mathrm{Lip}(f_{\boldsymbol{\theta}_{L+1}}, \mathcal{X}) := \sup_{\substack{\mathbf{x},\mathbf{y} \in \mathcal{X} \\ \mathbf{x} \neq \mathbf{y}}} \frac{\|f_{\boldsymbol{\theta}_{L+1}}(\mathbf{x}) - f_{\boldsymbol{\theta}_{L+1}}(\mathbf{y})\|_2}{\|\mathbf{x} - \mathbf{y}\|_2} \leq \sup_{\substack{\mathbf{x},\mathbf{y} \in \overline{\mathcal{B}}_{d_{\mathrm{in}}}(\mathbf{0}, \Lambda) \\ \mathbf{x} \neq \mathbf{y}}} \frac{\|f_{\boldsymbol{\theta}_{L+1}}(\mathbf{x}) - f_{\boldsymbol{\theta}_{L+1}}(\mathbf{y})\|_2}{\|\mathbf{x} - \mathbf{y}\|_2}. \tag{16}$$

Since the closed Euclidean ball is convex, the straight-line segment $\boldsymbol{\gamma}(t) := \mathbf{y} + t(\mathbf{x} - \mathbf{y})$, for any $\mathbf{x}, \mathbf{y} \in \overline{\mathcal{B}}_{d_{\mathrm{in}}}(\mathbf{0}, \Lambda)$ and $t \in [0, 1]$, lies entirely within it. Therefore, as the smoothness of the activation function ensures that the network $f_{\boldsymbol{\theta}_{L+1}}$ is continuously differentiable, the gradient theorem implies that

$$\left\|f_{\boldsymbol{\theta}_{L+1}}(\mathbf{x}) - f_{\boldsymbol{\theta}_{L+1}}(\mathbf{y})\right\|_2 = \left\|\int_0^1 J_{\boldsymbol{\theta}_{L+1}}\big(\boldsymbol{\gamma}(t)\big)\,\boldsymbol{\gamma}'(t)\,dt\right\|_2 \leq \|\mathbf{x} - \mathbf{y}\|_2 \int_0^1 \|J_{\boldsymbol{\theta}_{L+1}}\big(\boldsymbol{\gamma}(t)\big)\|\,dt$$

$$\leq \sup_{\mathbf{x}' \in \overline{\mathcal{B}}_{d_{\mathrm{in}}}(\mathbf{0}, \Lambda)} \|J_{\boldsymbol{\theta}_{L+1}}(\mathbf{x}')\|\,\|\mathbf{x} - \mathbf{y}\|_2.$$

The above inequality holds for any $\mathbf{x}, \mathbf{y} \in \mathcal{X}$, and thus it follows from Equation (16) that $\mathrm{Lip}(f_{\boldsymbol{\theta}_{L+1}}, \mathcal{X}) \leq \sup_{\mathbf{x} \in \overline{\mathcal{B}}_{d_{\mathrm{in}}}(\mathbf{0}, \Lambda)} \|J_{\boldsymbol{\theta}_{L+1}}(\mathbf{x})\|$, which proves inequality (8a). Finally, by noting that $J_{\boldsymbol{\theta}_{L+1}}(\mathbf{x}) = \sqrt{m} \cdot W_{L+1} J_{\boldsymbol{\theta}_L}(\mathbf{x})$, inequality (8b) also follows since

$$\mathrm{Lip}(f_{\boldsymbol{\theta}_{L+1}}, \mathcal{X}) \leq \sup_{\mathbf{x} \in \overline{\mathcal{B}}_{d_{\mathrm{in}}}(\mathbf{0}, \Lambda)} \sup_{\mathbf{y} \in \overline{\mathcal{B}}_{d_{\mathrm{in}}}(\mathbf{0}, 1)} \sup_{\mathbf{z} \in \overline{\mathcal{B}}_{d_{\mathrm{out}}}(\mathbf{0}, 1)} \left\langle \mathbf{z}, \sqrt{m} \cdot W_{L+1} J_{\boldsymbol{\theta}_L}(\mathbf{x})\mathbf{y}\right\rangle$$

$$= \sup_{(\mathbf{x},\mathbf{y},\mathbf{z}) \in \mathcal{T}} \left\langle \mathbf{z}, \sqrt{m} \cdot W_{L+1} J_{\boldsymbol{\theta}_L}(\mathbf{x})\mathbf{y}\right\rangle,$$

where we have used the fact that, for any matrix $A \in \mathbb{R}^{d_{\text{out}} \times d_{\text{in}}}$, its spectral norm can be written as $\|A\| = \sup_{\|\mathbf{y}\| \leq 1, \|\mathbf{z}\| \leq 1} \langle \mathbf{z}, A\mathbf{y} \rangle$. This concludes the proof.

Another complementary result concerns the behavior of the naive Lipschitz bound at initialization. We present its proof here.

### B.2. Proof of Proposition 3.8

We split the proof into two steps. First, we bound the spectral norm of each weight matrix at initialization. Then, we combine these bounds to control the naive Lipschitz bound.

**Step 1: Bounding the spectral norms.** For $\ell = 2, \ldots, L$, using Theorem 4.4.3 in (Vershynin, 2018), we have that for all $u \geq 0$

$$\|W_\ell\| \stackrel{d}{=} \frac{1}{\sqrt{m}} \|\bar{W}_\ell\| \leq \frac{1}{\sqrt{m}} (2\sqrt{m} + u) = 2 + \frac{u}{\sqrt{m}} \stackrel{(a)}{=} 3,$$

with probability at least $1 - 2\exp(-cm)$, where $[\bar{W}_\ell]_{ij} \stackrel{\text{i.i.d.}}{\sim} \mathcal{N}(0,1)$ for all $(i,j) \in [m] \times [m]$, and $c > 0$ is a universal constant. Equality (a) follows by setting $u = \sqrt{m}$. Similarly, using the assumption that $m \geq \max\{d_{\text{in}}, d_{\text{out}}\}$ we can show that $\|W_1\| \leq 3$ and $\|W_{L+1}\| \leq 3$, each with probability at least $1 - 2\exp(-cm)$.

**Step 2: Combining the bounds.** By the independence of the individual weight matrices, it follows that

$$\mathbb{P}\left(K_{\text{naive}}(\boldsymbol{\theta}_{L+1}) \leq \sqrt{m} \cdot 3^{L+1}\right) \geq \mathbb{P}\left(\forall \ell \in [L+1] : \|W_\ell\| \leq 3\right) = \prod_{\ell=1}^{L+1} \mathbb{P}\left(\|W_\ell \leq 3\|\right) \geq (1 - 2\exp(-cm))^{L+1},$$

which establishes the desired result.

### B.3. Lipschitz Bounds at Random Initialization

We begin by establishing high-probability bounds on the Lipschitz constant at random initialization. To this end, we first introduce the following definition.

**Definition B.1.** Let $W_1, \cdots, W_{L+1}$ be weight matrices initialized as in Assumption 3.5, and let $W_1, W_2, \ldots, W_L$ be fixed. The Lipschitz stochastic process $(V_t)_{t \in \mathcal{T}}$ over the randomness of $W_{L+1}$ is defined as

$$V_t := \langle \mathbf{z}, \sqrt{m} \cdot W_{L+1} J_{\boldsymbol{\theta}_L}(\mathbf{x})\mathbf{y} \rangle \quad \text{for all } t := (\mathbf{x}, \mathbf{y}, \mathbf{z}) \in \mathcal{T}, \tag{17}$$

where $\mathcal{T} := \overline{\mathcal{B}}_{d_{\text{in}}}(\mathbf{0}, \Lambda) \times \overline{\mathcal{B}}_{d_{\text{in}}}(\mathbf{0}, 1) \times \overline{\mathcal{B}}_{d_{\text{out}}}(\mathbf{0}, 1)$.

By Proposition 3.7, we know that the supremum of the Lipschitz process serves as an upper bound for the network's Lipschitz constant. Therefore, our goal is to derive an upper bound for this supremum, which we achieve by leveraging Dudley's integral. However, before applying Theorem A.6, we must first verify that the process $(V_t)_{t \in \mathcal{T}}$ satisfies all the necessary conditions for the theorem's application. The following lemma provides this verification.

**Lemma B.2.** *The Lipschitz process $(V_t)_{t \in \mathcal{T}}$ over the randomness of $W_{L+1}$ is subgaussian, i.e., the increments satisfy*

$$\|V_t - V_s\|_{\psi_2} \leq C\rho(t, s), \tag{18}$$

*for any $t := (\mathbf{x}, \mathbf{y}, \mathbf{z}) \in \mathcal{T}$ and $s := (\tilde{\mathbf{x}}, \tilde{\mathbf{y}}, \tilde{\mathbf{z}}) \in \mathcal{T}$, where $C > 0$ is a universal constant and the metric $\rho : \mathcal{T} \times \mathcal{T} \to \mathbb{R}_{\geq 0}$ is given by*

$$\rho(t, s) := \alpha_{\mathbf{x}} \|\mathbf{x} - \tilde{\mathbf{x}}\|_2 + \alpha_{\mathbf{y}, \mathbf{z}} \|\mathbf{y} - \tilde{\mathbf{y}}\|_2 + \alpha_{\mathbf{y}, \mathbf{z}} \|\mathbf{z} - \tilde{\mathbf{z}}\|_2, \tag{19}$$

*with*

$$\alpha_{\mathbf{x}} := \beta_\phi \prod_{i=1}^{L} \|W_i\| \sum_{r=1}^{L} \|W_r\|_{2 \to \infty} \prod_{j=1}^{r-1} \|W_j\|,$$

*and*

$$\alpha_{\mathbf{y}, \mathbf{z}} := \prod_{i=1}^{L} \|W_i\|.$$

*Proof.* It is easy to see that, conditioned on $W_1, \ldots, W_L$, for any $t, s \in \mathcal{T}$, the increments $V_t - V_s$ are Gaussian, and therefore possess a finite subgaussian norm. To analyze this norm, we decompose the increments in three parts and treat each part separately. We have that

$$
\begin{aligned}
\|V_t - V_s\|_{\psi_2} = \sqrt{m} \, \|\langle \mathbf{z}, W_{L+1} J_{\boldsymbol{\theta}_L}(\mathbf{x})\mathbf{y}\rangle - \langle \tilde{\mathbf{z}}, W_{L+1} J_{\boldsymbol{\theta}_L}(\tilde{\mathbf{x}})\tilde{\mathbf{y}}\rangle\|_{\psi_2} \\
\leq \sqrt{m} \, \underbrace{\|\langle \mathbf{z}, W_{L+1} J_{\boldsymbol{\theta}_L}(\mathbf{x})\mathbf{y}\rangle - \langle \tilde{\mathbf{z}}, W_{L+1} J_{\boldsymbol{\theta}_L}(\mathbf{x})\mathbf{y}\rangle\|_{\psi_2}}_{=:M_1} \\
+ \sqrt{m} \, \underbrace{\|\langle \tilde{\mathbf{z}}, W_{L+1} J_{\boldsymbol{\theta}_L}(\mathbf{x})\mathbf{y}\rangle - \langle \tilde{\mathbf{z}}, W_{L+1} J_{\boldsymbol{\theta}_L}(\tilde{\mathbf{x}})\mathbf{y}\rangle\|_{\psi_2}}_{=:M_2} \\
+ \sqrt{m} \, \underbrace{\|\langle \tilde{\mathbf{z}}, W_{L+1} J_{\boldsymbol{\theta}_L}(\tilde{\mathbf{x}})\mathbf{y}\rangle - \langle \tilde{\mathbf{z}}, W_{L+1} J_{\boldsymbol{\theta}_L}(\tilde{\mathbf{x}})\tilde{\mathbf{y}}\rangle\|_{\psi_2}}_{=:M_3}
\end{aligned}
\tag{20}
$$

where we have applied the triangle inequality for the subgaussian norm $\|\cdot\|_{\psi_2}$.

**First term in Equation** (20): For the term $M_1$ we have

$$
\begin{aligned}
M_1 &= \|\langle \mathbf{z} - \tilde{\mathbf{z}}, W_{L+1} J_{\boldsymbol{\theta}_L}(\mathbf{x})\mathbf{y}\rangle\|_{\psi_2} \\
&= \left\| \sum_{r=1}^{d_{\text{out}}} (z_r - \tilde{z}_r) \left\langle \mathbf{w}_{L+1}^{(r)}, J_{\boldsymbol{\theta}_L}(\mathbf{x})\mathbf{y}\right\rangle \right\|_{\psi_2} \\
&\overset{(a)}{\lesssim} \left( \sum_{r=1}^{d_{\text{out}}} \left\| (z_r - \tilde{z}_r) \left\langle \mathbf{w}_{L+1}^{(r)}, J_{\boldsymbol{\theta}_L}(\mathbf{x})\mathbf{y}\right\rangle \right\|_{\psi_2}^2 \right)^{\frac{1}{2}} \\
&\overset{(b)}{\lesssim} \frac{1}{\sqrt{m}} \|J_{\boldsymbol{\theta}_L}(\mathbf{x})\mathbf{y}\|_2 \left( \sum_{r=1}^{d_{\text{out}}} |z_r - \tilde{z}_r|^2 \right)^{\frac{1}{2}} \\
&\overset{(c)}{\leq} \frac{1}{\sqrt{m}} \prod_{i=1}^{L} \|W_i\| \, \|\mathbf{z} - \tilde{\mathbf{z}}\|_2 = \frac{\alpha_{\mathbf{y}, \mathbf{z}}}{\sqrt{m}} \|\mathbf{z} - \tilde{\mathbf{z}}\|_2,
\end{aligned}
$$

where $\mathbf{w}_{L+1}^{(r)} \in \mathbb{R}^m$ denote the rows of the matrix $W_{L+1}$. Inequality (a) follows from Proposition A.2 since the variables $\left\langle \mathbf{w}_{L+1}^{(r)}, J_{\boldsymbol{\theta}_L}(\mathbf{x})\mathbf{y}\right\rangle \overset{\text{i.i.d.}}{\sim} \mathcal{N}\left(0, m^{-1} \|J_{\boldsymbol{\theta}_L}(\mathbf{x})\mathbf{y}\|_2^2\right)$ are independent, mean-zero, Gaussian random variables for all $r \in [d_{\text{out}}]$. Inequality (b) follows from Exercise 2.24 in (Vershynin, 2018), and inequality (c) is a direct consequence of the 1-Lipschitz property of the activation function, and the fact that $\mathbf{y} \in \overline{\mathcal{B}}_{d_{\text{in}}}(\mathbf{0}, 1)$.

**Second term in Equation** (20): Similarly, a bound for $M_2$ can be established in the following way

$$M_2 = \|\langle \tilde{\mathbf{z}}, W_{L+1} [J_{\boldsymbol{\theta}_L}(\mathbf{x}) - J_{\boldsymbol{\theta}_L}(\tilde{\mathbf{x}})] \mathbf{y} \rangle\|_{\psi_2}$$

$$= \left\| \sum_{r=1}^{d_{\text{out}}} \tilde{z}_r \left\langle \mathbf{w}_{L+1}^{(r)}, [J_{\boldsymbol{\theta}_L}(\mathbf{x}) - J_{\boldsymbol{\theta}_L}(\tilde{\mathbf{x}})] \mathbf{y} \right\rangle \right\|_{\psi_2}$$

$$\overset{(a)}{\lesssim} \left( \sum_{r=1}^{d_{\text{out}}} \left\| \tilde{z}_r \left\langle \mathbf{w}_{L+1}^{(r)}, [J_{\boldsymbol{\theta}_L}(\mathbf{x}) - J_{\boldsymbol{\theta}_L}(\tilde{\mathbf{x}})] \mathbf{y} \right\rangle \right\|_{\psi_2}^2 \right)^{\frac{1}{2}}$$

$$\overset{(b)}{\lesssim} \frac{1}{\sqrt{m}} \left( \sum_{r=1}^{d_{\text{out}}} |\tilde{z}_r|^2 \right)^{\frac{1}{2}} \|[J_{\boldsymbol{\theta}_L}(\mathbf{x}) - J_{\boldsymbol{\theta}_L}(\tilde{\mathbf{x}})] \mathbf{y}\|_2$$

$$\overset{(c)}{\leq} \frac{1}{\sqrt{m}} \left( \beta_\phi \prod_{i=1}^{L} \|W_i\| \sum_{r=1}^{L} \|W_r\|_{2\to\infty} \prod_{j=1}^{r-1} \|W_j\| \right) \|\mathbf{x} - \tilde{\mathbf{x}}\|_2 = \frac{\alpha_{\mathbf{x}}}{\sqrt{m}} \|\mathbf{x} - \tilde{\mathbf{x}}\|_2.$$

where (a) and (b) follow from Proposition A.2, and Exercise 2.24 in (Vershynin, 2018) respectively, and (c) follows from Lemma C.2, and by noting that $\tilde{\mathbf{z}} \in \overline{\mathcal{B}}_{d_{\text{out}}}(\mathbf{0}, 1)$ and $\mathbf{y} \in \overline{\mathcal{B}}_{d_{\text{in}}}(\mathbf{0}, 1)$.

**Third term in Equation** (20): For the final term $M_3$, we obtain

$$M_3 = \|\langle \tilde{\mathbf{z}}, W_{L+1} J_{\boldsymbol{\theta}_L}(\tilde{\mathbf{x}})(\mathbf{y} - \tilde{\mathbf{y}}) \rangle\|_{\psi_2}$$

$$= \left\| \sum_{r=1}^{d_{\text{out}}} \tilde{z}_r \left\langle \mathbf{w}_{L+1}^{(r)}, J_{\boldsymbol{\theta}_L}(\tilde{\mathbf{x}})(\mathbf{y} - \tilde{\mathbf{y}}) \right\rangle \right\|_{\psi_2}$$

$$\overset{(a)}{\lesssim} \left( \sum_{r=1}^{d_{\text{out}}} \left\| \tilde{z}_r \left\langle \mathbf{w}_{L+1}^{(r)}, J_{\boldsymbol{\theta}_L}(\tilde{\mathbf{x}})(\mathbf{y} - \tilde{\mathbf{y}}) \right\rangle \right\|_{\psi_2}^2 \right)^{\frac{1}{2}}$$

$$\overset{(b)}{\lesssim} \frac{1}{\sqrt{m}} \left( \sum_{r=1}^{d_{\text{out}}} |\tilde{z}_r|^2 \right)^{\frac{1}{2}} \|J_{\boldsymbol{\theta}_L}(\tilde{\mathbf{x}})(\mathbf{y} - \tilde{\mathbf{y}})\|_2$$

$$\overset{(c)}{\leq} \frac{1}{\sqrt{m}} \prod_{i=1}^{L} \|W_i\| \|\mathbf{y} - \tilde{\mathbf{y}}\|_2 = \frac{\alpha_{\mathbf{y},\mathbf{z}}}{\sqrt{m}} \|\mathbf{y} - \tilde{\mathbf{y}}\|_2.$$

where (a) and (b) follow from Proposition A.2, and Exercise 2.24 in (Vershynin, 2018), and (c) follows from the 1-Lipschitz property of the activation. Finally, combining the bounds for $M_1$, $M_2$ and $M_3$ and substituting into Equation (20) we derive the desired result. $\square$

Lemma B.2 verifies that the process $(V_t)_{t\in\mathcal{T}}$ has subgaussian increments, and hence, Dudley's inequality can be applied. The subsequent lemma then provides an explicit upper bound on the metric entropy of the set $\mathcal{T}$.

**Lemma B.3.** *Let $\mathcal{T}$ be defined as in B.1. Then, if the weight matrices $W_1, \ldots, W_L$ are fixed, for any $u \geq 0$, the Lipschitz process $(V_t)_{t\in\mathcal{T}}$ satisfies*

$$\sup_{t\in\mathcal{T}} V_t \leq C \left[ (\Lambda \alpha_{\mathbf{x}} + \alpha_{\mathbf{y},\mathbf{z}}) \left( \sqrt{d_{\text{in}}} + u \right) + \alpha_{\mathbf{y},\mathbf{z}} \sqrt{d_{\text{out}}} \right], \tag{21}$$

*with probability at least $1 - 2\exp(-u^2)$ over the randomness of $W_{L+1}$, where $C > 0$ is a universal constant.*

*Proof.* To establish the proof, we will make use of Dudley's inequality as expressed in Equation (15). We first note that by Lemma B.2, the Lipschitz process $(V_t)_{t\in\mathcal{T}}$ is subgaussian with respect to the metric of Equation (19). Additionally, it is

easy to verify that $\mathbb{E}[V_t] = 0$, for all $t \in \mathcal{T}$. Thus, the conditions to apply Dudley's integral inequality hold, and by Theorem A.6, we know that for any $u \geq 0$, the event

$$\sup_{t,s\in\mathcal{T}} |V_t - V_s| \leq C_1 \left[ \int_0^\infty \sqrt{\log N(\mathcal{T}, \rho, \varepsilon)} \, d\varepsilon + u \cdot \text{diam}(\mathcal{T}) \right], \tag{22}$$

holds with probability at least $1 - 2\exp(-u^2)$, where $C_1 > 0$ is an absolute constant, and $\text{diam}(\mathcal{T}) := \sup_{t,s\in\mathcal{T}} \rho(t,s)$ denotes the diameter of the set $\mathcal{T}$. We will proceed by bounding the covering number of the index set $\mathcal{T}$. To this end, in order to simplify the notation, we define the quantities

$$N_{\mathbf{x}}(\varepsilon) := N\left( \overline{\mathcal{B}}_{d_{\text{in}}}(\mathbf{0}, \Lambda), \ \|\cdot\|_2, \ \frac{\varepsilon}{3\alpha_{\mathbf{x}}} \right),$$

$$N_{\mathbf{y}}(\varepsilon) := N\left( \overline{\mathcal{B}}_{d_{\text{in}}}(\mathbf{0}, 1), \ \|\cdot\|_2, \ \frac{\varepsilon}{3\alpha_{\mathbf{y},\mathbf{z}}} \right),$$

$$N_{\mathbf{z}}(\varepsilon) := N\left( \overline{\mathcal{B}}_{d_{\text{out}}}(\mathbf{0}, 1), \ \|\cdot\|_2, \ \frac{\varepsilon}{3\alpha_{\mathbf{y},\mathbf{z}}} \right)$$

By Lemma D.2, we know that $N(\mathcal{T}, \rho, \varepsilon) \leq N_{\mathbf{x}}(\varepsilon) \cdot N_{\mathbf{y}}(\varepsilon) \cdot N_{\mathbf{z}}(\varepsilon)$, where the metric $\rho(\cdot)$ is defined as in Equation (19). Hence, for any $\varepsilon > 0$, the metric entropy of $\mathcal{T}$ satisfies $\log N(\mathcal{T}, \rho, \varepsilon) \leq \log N_{\mathbf{x}}(\varepsilon) + \log N_{\mathbf{y}}(\varepsilon) + \log N_{\mathbf{z}}(\varepsilon)$. By defining $M := \int_0^\infty \sqrt{\log N(\mathcal{T}, \rho, \varepsilon)} \, d\varepsilon$ and using the elementary inequality $\sqrt{x+y} \leq \sqrt{x} + \sqrt{y}$, which is valid for any $x, y \geq 0$, we get

$$M \leq \int_0^\infty \sqrt{\log N_{\mathbf{x}}(\varepsilon)} \, d\varepsilon + \int_0^\infty \sqrt{\log N_{\mathbf{y}}(\varepsilon)} \, d\varepsilon + \int_0^\infty \sqrt{\log N_{\mathbf{z}}(\varepsilon)} \, d\varepsilon.$$

Moreover, we observe that the set $\overline{\mathcal{B}}_{d_{\text{in}}}(\mathbf{0}, \Lambda)$ can be covered by a a single element whenever $\varepsilon > 3\Lambda\alpha_{\mathbf{x}}$, which implies that $\log N_{\mathbf{x}}(\varepsilon) = 0$ in that range. By the same reasoning, we deduce that $\log N_{\mathbf{y}}(\varepsilon) = \log N_{\mathbf{z}}(\varepsilon) = 0$ for all $\varepsilon > 3\alpha_{\mathbf{y},\mathbf{z}}$. It follows that

$$M \leq \int_0^{3\Lambda\alpha_{\mathbf{x}}} \sqrt{\log N_{\mathbf{x}}(\varepsilon)} \, d\varepsilon + \int_0^{3\alpha_{\mathbf{y},\mathbf{z}}} \sqrt{\log N_{\mathbf{y}}(\varepsilon)} \, d\varepsilon + \int_0^{3\alpha_{\mathbf{y},\mathbf{z}}} \sqrt{\log N_{\mathbf{z}}(\varepsilon)} \, d\varepsilon. \tag{23}$$

By Proposition A.5 we can derive the following upper bounds

$$N_{\mathbf{x}}(\varepsilon) \leq \left( 1 + \frac{6\Lambda\alpha_{\mathbf{x}}}{\varepsilon} \right)^{d_{\text{in}}} \leq \left( \frac{9\Lambda\alpha_{\mathbf{x}}}{\varepsilon} \right)^{d_{\text{in}}} \qquad \text{for all } 0 < \varepsilon \leq 3\Lambda\alpha_{\mathbf{x}},$$

$$N_{\mathbf{y}}(\varepsilon) \leq \left( 1 + \frac{6\alpha_{\mathbf{y},\mathbf{z}}}{\varepsilon} \right)^{d_{\text{in}}} \leq \left( \frac{9\alpha_{\mathbf{y},\mathbf{z}}}{\varepsilon} \right)^{d_{\text{in}}} \qquad \text{for all } 0 < \varepsilon \leq 3\alpha_{\mathbf{y},\mathbf{z}}, \tag{24}$$

$$N_{\mathbf{z}}(\varepsilon) \leq \left( 1 + \frac{6\alpha_{\mathbf{y},\mathbf{z}}}{\varepsilon} \right)^{d_{\text{out}}} \leq \left( \frac{9\alpha_{\mathbf{y},\mathbf{z}}}{\varepsilon} \right)^{d_{\text{out}}} \qquad \text{for all } 0 < \varepsilon \leq 3\alpha_{\mathbf{y},\mathbf{z}}.$$

Substituting the inequalities of Equation (24) into Equation (23), we get

$$M \leq \sqrt{d_{\text{in}}} \int_0^{3\Lambda\alpha_{\mathbf{x}}} \sqrt{\log\left( \frac{9\Lambda\alpha_{\mathbf{x}}}{\varepsilon} \right)} \, d\varepsilon + (\sqrt{d_{\text{in}}} + \sqrt{d_{\text{out}}}) \int_0^{3\alpha_{\mathbf{y},\mathbf{z}}} \sqrt{\log\left( \frac{9\alpha_{\mathbf{y},\mathbf{z}}}{\varepsilon} \right)} \, d\varepsilon$$

$$= 9\Lambda\alpha_{\mathbf{x}} \sqrt{d_{\text{in}}} \int_0^{\frac{1}{3}} \sqrt{\log\left( \frac{1}{\sigma} \right)} \, d\sigma + 9\alpha_{\mathbf{y},\mathbf{z}} (\sqrt{d_{\text{in}}} + \sqrt{d_{\text{out}}}) \int_0^{\frac{1}{3}} \sqrt{\log\left( \frac{1}{\sigma} \right)} \, d\sigma$$

$$= C_2 \left[ (\Lambda\alpha_{\mathbf{x}} + \alpha_{\mathbf{y},\mathbf{z}}) \sqrt{d_{\text{in}}} + \alpha_{\mathbf{y},\mathbf{z}} \sqrt{d_{\text{out}}} \right], \tag{25}$$

where we defined $C_2 := 9 \int_0^{1/3} \sqrt{\log\left( \frac{1}{\sigma} \right)} \, d\sigma$. Finally, by noting that $\text{diam}(\mathcal{T}) = 2\Lambda\alpha_{\mathbf{x}} + 4\alpha_{\mathbf{y},\mathbf{z}} < 4\Lambda\alpha_{\mathbf{x}} + 4\alpha_{\mathbf{y},\mathbf{z}}$ it follows

that

$$\sup_{t,s\in\mathcal{T}} |V_t - V_s| \overset{(a)}{\leq} C_1 \left\{ C_2 \left[ (\Lambda\alpha_{\mathbf{x}} + \alpha_{\mathbf{y},\mathbf{z}})\sqrt{d_{\text{in}}} + \alpha_{\mathbf{y},\mathbf{z}}\sqrt{d_{\text{out}}} \right] + (2\Lambda\alpha_{\mathbf{x}} + 4\alpha_{\mathbf{y},\mathbf{z}})u \right\}$$

$$< C_1 \max\{4, C_2\} \left[ (\Lambda\alpha_{\mathbf{x}} + \alpha_{\mathbf{y},\mathbf{z}})(\sqrt{d_{\text{in}}} + u) + \alpha_{\mathbf{y},\mathbf{z}}\sqrt{d_{\text{out}}} \right]$$

$$\overset{(b)}{\leq} C \left[ (\Lambda\alpha_{\mathbf{x}} + \alpha_{\mathbf{y},\mathbf{z}})(\sqrt{d_{\text{in}}} + u) + \alpha_{\mathbf{y},\mathbf{z}}\sqrt{d_{\text{out}}} \right],$$

where (a) follows by combining Equation (22) and (25), and (b) by defining $C := C_1 \max\{4, C_2\}$. The proof is concluded by observing that for $t_0 := (\mathbf{0}, \mathbf{0}, \mathbf{0}) \in \mathcal{T}$, the Lipschitz process satisfies $V_{t_0} = 0$ a.s., and that

$$\sup_{t\in\mathcal{T}} V_t \leq \sup_{t\in\mathcal{T}} |V_t - V_{t_0}| \leq \sup_{t,s\in\mathcal{T}} |V_t - V_s|$$

$\square$

We have now established a bound for the supremum of the process $(V_t)_{t\in\mathcal{T}}$ with respect to the randomness of the final weight matrix $W_{L+1}$, while considering the remaining weights as fixed. The following lemmas aim to reintroduce the randomness of $W_1, \ldots, W_L$, which influence the coefficients $\alpha_{\mathbf{x}}$, and $\alpha_{\mathbf{y},\mathbf{z}}$. To this end, we begin with the following observation.

**Lemma B.4.** *Let the matrices $W_1, \ldots, W_L$ be initialized as in Assumption 3.5. We define the events*

$$\mathcal{W}_{\text{op}} := \left\{ (W_1, \ldots, W_L) \ : \ \|W_\ell\| \leq 2 + \frac{2}{L}, \ \forall \ell \in [L] \right\}, \tag{26}$$

$$\mathcal{W}_{2\to\infty} := \left\{ (W_1, \ldots, W_L) \ : \ \|W_\ell\|_{2\to\infty} \leq 1 + \frac{1}{L}, \ \forall \ell \in [L] \right\}. \tag{27}$$

*The event $\mathcal{W}_{\text{op}}$ jointly bounds the operator norms of all hidden layer weight matrices, whereas $\mathcal{W}_{2\to\infty}$ bounds their two-to-infinity norms. If the networks width satisfies $m \geq \max\{d_{\text{in}}, CL^2 \log m\}$, we have the following estimate*

$$\mathbb{P}\left(\mathcal{W}_{\text{op}} \cap \mathcal{W}_{2\to\infty}\right) \geq \left(1 - 4\exp(-cm/L^2)\right)^L, \tag{28}$$

*where $C, c > 0$ are universal constants.*

*Proof.* In the proof, we will use the notation $X \overset{d}{=} Y$ to indicate equality in distribution between two random variables $X$ and $Y$. We split the proof in three steps. In the first and second step, we will establish bounds for the spectral and two-to-infinity norm of each individual weight matrix, respectively. Then, we will combine these bounds to derive the inequality of Equation (28).

**Step 1: Bounding the spectral norms.** We have that for any $u \geq 0$ and for $\ell = 2, \ldots, L$,

$$\|W_\ell\| \overset{d}{=} \frac{1}{\sqrt{m}}\|\bar{W}_\ell\| \overset{(a)}{\leq} \frac{1}{\sqrt{m}}(2\sqrt{m} + u) = 2 + \frac{u}{\sqrt{m}} \overset{(b)}{=} 2 + \frac{2}{L},$$

holds with probability at least $1 - 2\exp(-c_1 m/L^2)$, where $[\bar{W}_\ell]_{i,j} \overset{\text{i.i.d.}}{\sim} \mathcal{N}(0,1)$ for all $(i,j) \in [m] \times [m]$, and $c_1 > 0$ is an absolute constant. Inequality (a) follows from Theorem 4.4.3 in (Vershynin, 2018) while equality (b) holds by choosing $u = 2\sqrt{m}/L$. Similarly, for $\ell = 1$ we have that

$$\|W_1\| \overset{d}{=} \frac{1}{\sqrt{m}}\|\bar{W}_1\| \leq \frac{1}{\sqrt{m}}(\sqrt{d_{\text{in}}} + \sqrt{m} + u) \overset{(c)}{\leq} 2 + \frac{2}{L},$$

holds with probability $1 - 2\exp(-c_1 m/L^2)$, where $[\bar{W}_1]_{i,j} \overset{\text{i.i.d.}}{\sim} \mathcal{N}(0,1)$ for $(i,j) \in [m] \times [d_{\text{in}}]$. Here, (c) follows from the assumption that $m \geq d_{\text{in}}$.

**Step 2: Bounding the two-to-infinity norms.** For any $u > 0$ and for $\ell = 2, \ldots, L$,

$$\|W_\ell\|_{2\to\infty} \overset{d}{=} \frac{1}{\sqrt{m}}\|\bar{W}_\ell\|_{2\to\infty} \overset{(a)}{\leq} \frac{1}{\sqrt{m}}(\sqrt{m} + \tilde{C}\sqrt{\log m} + u) \overset{(b)}{\leq} 1 + \frac{1}{L},$$

holds with probability at least $1 - 2\exp(-c_2 m/L^2)$, where $\tilde{C}, c_2 > 0$ are absolute constants. Inequality (a) follows from Lemma D.1, while (b) follows by setting $u = \sqrt{m}/2L$, and noting that by assumption $m \geq CL^2 \log m$, with $C := 4\tilde{C}^2$. Moreover, for $\ell = 1$ we get that

$$\|W_1\|_{2\to\infty} \overset{d}{=} \frac{1}{\sqrt{m}}\|\bar{W}_1\|_{2\to\infty} \leq \frac{1}{\sqrt{m}}(\sqrt{d_{\text{in}}} + \tilde{C}\sqrt{\log m} + u) \overset{(c)}{\leq} 1 + \frac{1}{L},$$

holds with probability at least $1 - 2\exp(-c_2 m/L^2)$, where like before, inequality (c) follows from the assumption that $m \geq \max\{d_{\text{in}}, CL^2 \log m\}$.

**Step 3: Combining the bounds:** By the union bound and taking advantage of the independence of the weight matrices at different layers, we obtain the following bound for the probability of the events $\mathcal{W}_{\text{op}}$ and $\mathcal{W}_{2\to\infty}$

$$\mathbb{P}\left(\mathcal{W}_{\text{op}} \cap \mathcal{W}_{2\to\infty}\right) \geq \left(1 - 4\exp(-cm/L^2)\right)^L, \tag{29}$$

where we have defined the constant $c := \min\{c_1, c_2\}$, which concludes the proof. $\qquad\square$

**Lemma B.5.** *Let $W_1, \ldots, W_L$ be a collection of weight matrices and let $\alpha_{\mathbf{x}}$ and $\alpha_{\mathbf{y},\mathbf{z}}$ be defined as in Lemma B.2. Then, we have the following implication*

$$(W_1, \ldots W_L) \in \mathcal{W}_{\text{op}} \cap \mathcal{W}_{2\to\infty} \implies \begin{cases} \alpha_{\mathbf{x}} \leq e^2 \beta_\phi \, 4^L, \\ \alpha_{\mathbf{y},\mathbf{z}} \leq e\, 2^L \end{cases} \tag{30}$$

*Proof.* We will bound the two terms separately. For the term $\alpha_{\mathbf{x}}$ we have that

$$\alpha_{\mathbf{x}} = \beta_\phi \left(\prod_{i=1}^L \|W_i\|\right)\left(\sum_{r=1}^L \|W_r\|_{2\to\infty} \prod_{j=1}^{r-1}\|W_j\|\right)$$

$$\overset{(\mathcal{W}_{\text{op}}, \mathcal{W}_{2\to\infty})}{\leq} \beta_\phi \left(\prod_{i=1}^L \left(2 + \frac{2}{L}\right)\right)\left(\sum_{r=1}^L \left(1 + \frac{1}{L}\right)\prod_{j=1}^{r-1}\left(2 + \frac{2}{L}\right)\right)$$

$$= \beta_\phi \left(1 + \frac{1}{L}\right)^L 2^L \sum_{r=1}^L 2^{r-1}\left(1 + \frac{1}{L}\right)^r$$

$$\leq \beta_\phi 4^L \left(1 + \frac{1}{L}\right)^{2L} \sum_{k=1}^L 2^{-k}$$

$$\overset{(a)}{\leq} \beta_\phi 4^L e^2 \sum_{k=1}^\infty 2^{-k}$$

$$\overset{(b)}{=} e^2 \beta_\phi \, 4^L.$$

Here, step (a) uses the inequality $(1 + \frac{1}{L})^L \leq e$, valid for all natural numbers $L$, and step (b) relies on the fact that the geometric series $\sum_{k=1}^\infty 2^{-k}$ converges to 1. Similarly, for the term $\alpha_{\mathbf{y},\mathbf{z}}$ we have

$$\alpha_{\mathbf{y},\mathbf{z}} = \prod_{i=1}^L \|W_i\| \overset{(\mathcal{W}_{\text{op}}, \mathcal{W}_{2\to\infty})}{\leq} \prod_{i=1}^L \left(2 + \frac{2}{L}\right) \leq \left(1 + \frac{1}{L}\right)^L 2^L \leq e\, 2^L.$$

which concludes the proof. $\qquad\square$

Having stated and proved all the necessary lemmas, we proceed with the proof of Theorem 4.1 establishing the Lipschitz bound during random initialization.

### B.4. Proof of Theorem 4.1

We start by defining the following three sets

$$\mathcal{A}_1 := \left\{ (W_1, \ldots, W_L) : \alpha_{\mathbf{x}} \le e^2 \beta_\phi \, 4^L \text{ and, } \alpha_{\mathbf{y,z}} \le e \, 2^L \right\},$$

$$\mathcal{A}_2(W_1, \ldots, W_L) := \left\{ W_{L+1} : \sup_{t \in \mathcal{T}} V_t \le \widetilde{C} \left[ (\Lambda \alpha_{\mathbf{x}} + \alpha_{\mathbf{y,z}})(\sqrt{d_{\text{in}}} + u) + \alpha_{\mathbf{y,z}} \sqrt{d_{\text{out}}} \right] \right\},$$

$$\mathcal{A} := \left\{ (W_1, \ldots, W_{L+1}) : \sup_{t \in \mathcal{T}} V_t \le C \, 2^L \left[ \left( e \, 2^L \beta_\phi \, \Lambda + 1 \right) (\sqrt{d_{\text{in}}} + u) + \sqrt{d_{\text{out}}} \right] \right\}$$

We will first show that if $(W_1, \ldots, W_L) \in \mathcal{A}_1$ and $W_{L+1} \in \mathcal{A}_2(W_1, \ldots, W_{L+1})$, then $(W_1, \ldots W_{L+1}) \in \mathcal{A}$. This statement follows immediately since

$$\sup_{t \in T} V_t \le \tilde{C} \left[ (e^2 \beta_\phi \Lambda \, 4^L + e \, 2^L)(\sqrt{d_{\text{in}}} + u) + e \, 2^L \sqrt{d_{\text{out}}} \right]$$

$$\le C \, 2^L \left[ \left( e \, 2^L \beta_\phi \, \Lambda + 1 \right) (\sqrt{d_{\text{in}}} + u) + \sqrt{d_{\text{out}}} \right],$$

where we have defined $C := e \, \widetilde{C}$. Hence, by combining Lemma B.3 and Proposition C.1 in (Geuchen et al., 2025), it follows that

$$\mathbb{P}(\mathcal{A}) \ge \mathbb{P}(\mathcal{A}_1) \inf_{(W_1, \ldots, W_L) \in \mathcal{A}_1} \mathbb{P}\big(\mathcal{A}_2(W_1, \ldots, W_L)\big)$$

$$\ge \mathbb{P}(\mathcal{A}_1)\big(1 - 2\exp(-u^2)\big),$$

To establish the final result, we note that Lemma B.4 and Lemma B.5 suggest that

$$\mathbb{P}(\mathcal{A}_1) \ge \mathbb{P}(\mathcal{W}_{\text{op}} \cap \mathcal{W}_{2 \to \infty}) \ge \big(1 - 4\exp(-cm/L^2)\big)^L.$$

This concludes the proof.

### B.5. Lipschitz Bounds During Lazy Training

With the proof of Theorem 4.1 in place, we now turn to bounding the perturbation of the Lipschitz constant, uniformly over the lazy training function class $\mathcal{F}(\boldsymbol{\theta}_{L+1}, R)$. We start by making the following observation.

**Lemma B.6.** *Let $W_1, \ldots, W_{L+1}$ and $\widetilde{W}_1, \ldots, \widetilde{W}_{L+1}$ be two sets of network parameters. Then, the following inequality is satisfied*

$$\text{Lip}(f_{\tilde{\boldsymbol{\theta}}_{L+1}}, \mathcal{X}) \le \sup_{t \in \mathcal{T}} V_t + \sqrt{m} \, \gamma(\boldsymbol{\theta}_L, \tilde{\boldsymbol{\theta}}_L) \, \|W_{L+1}\| + \sqrt{m} \, \|\widetilde{W}_{L+1} - W_{L+1}\| \prod_{\ell=1}^{L} \|\widetilde{W}_\ell\|, \tag{31}$$

*where*

$$\gamma(\boldsymbol{\theta}_L, \tilde{\boldsymbol{\theta}}_L) := \sum_{r=1}^{L} \left( \prod_{i=r+1}^{L} \|W_i\| \right) \left( \beta_\phi \Lambda \sum_{k=1}^{L-r+1} \prod_{p=1}^{k+r-1} \|\widetilde{W}_p\| + 1 \right) \left( \prod_{j=1}^{r-1} \|\widetilde{W}_j\| \right) \|\widetilde{W}_r - W_r\|,$$

*and $f_{\tilde{\boldsymbol{\theta}}_{L+1}}$ is the neural network defined as in Equation (3).*

*Proof.* First, we establish the following uniform bound for the Jacobian matrix perturbation in the following way

$$\sup_{\mathbf{x} \in \overline{\mathcal{B}}_{d_{\mathrm{in}}}(\mathbf{0},\Lambda)} \|J_{\tilde{\boldsymbol{\theta}}_{L+1}}(\mathbf{x}) - J_{\boldsymbol{\theta}_{L+1}}(\mathbf{x})\| = \sup_{\mathbf{x} \in \overline{\mathcal{B}}_{d_{\mathrm{in}}}(\mathbf{0},\Lambda)} \|\sqrt{m} \cdot \widetilde{W}_{L+1} J_{\tilde{\boldsymbol{\theta}}_L}(\mathbf{x}) - \sqrt{m} \cdot W_{L+1} J_{\boldsymbol{\theta}_L}(\mathbf{x})\|$$

$$\leq \sup_{\mathbf{x} \in \overline{\mathcal{B}}_{d_{\mathrm{in}}}(\mathbf{0},\Lambda)} \left\{ \sqrt{m} \|W_{L+1} \Delta J(\mathbf{x})\| + \sqrt{m} \|\widetilde{W}_{L+1} - W_{L+1}\| \|J_{\tilde{\boldsymbol{\theta}}_L}(\mathbf{x})\| \right\}$$

$$\leq \sup_{\mathbf{x} \in \overline{\mathcal{B}}_{d_{\mathrm{in}}}(\mathbf{0},\Lambda)} \left\{ \sqrt{m} \|\Delta J(\mathbf{x})\| \|W_{L+1}\| \right\} + \sqrt{m} \|\widetilde{W}_{L+1} - W_{L+1}\| \prod_{\ell=1}^{L} \|\widetilde{W}_\ell\|$$

$$\leq \sqrt{m} \gamma(\boldsymbol{\theta}_L, \tilde{\boldsymbol{\theta}}_L) \|W_{L+1}\| + \sqrt{m} \|\widetilde{W}_{L+1} - W_{L+1}\| \prod_{\ell=1}^{L} \|\widetilde{W}_\ell\| \tag{32}$$

where we have defined $\Delta J(\mathbf{x}) := J_{\tilde{\boldsymbol{\theta}}_L}(\mathbf{x}) - J_{\boldsymbol{\theta}_L}(\mathbf{x})$, and the final inequality follows from Lemma C.3. Now consider a fixed $\mathbf{x} \in \overline{\mathcal{B}}_{d_{\mathrm{in}}}(\mathbf{0}, \Lambda)$. A simple application of the triangle inequality yields

$$\|J_{\tilde{\boldsymbol{\theta}}_{L+1}}(\mathbf{x})\| \leq \|J_{\boldsymbol{\theta}_{L+1}}(\mathbf{x})\| + \|J_{\tilde{\boldsymbol{\theta}}_{L+1}}(\mathbf{x}) - J_{\boldsymbol{\theta}_{L+1}}(\mathbf{x})\| \leq \sqrt{m} \|W_{L+1} J_{\boldsymbol{\theta}_L}(\mathbf{x})\| + \|J_{\tilde{\boldsymbol{\theta}}_{L+1}}(\mathbf{x}) - J_{\boldsymbol{\theta}_{L+1}}(\mathbf{x})\|.$$

Since the above inequality holds for all $\mathbf{x} \in \overline{\mathcal{B}}_{d_{\mathrm{in}}}(\mathbf{0}, \Lambda)$, by taking the supremum on both sides and substituting the bound of Equation (32), we get

$$\sup_{\mathbf{x} \in \overline{\mathcal{B}}_{d_{\mathrm{in}}}(\mathbf{0},\Lambda)} \|J_{\tilde{\boldsymbol{\theta}}_{L+1}}(\mathbf{x})\| \leq \sup_{\mathbf{x} \in \overline{\mathcal{B}}_{d_{\mathrm{in}}}(\mathbf{0},\Lambda)} \left\{ \sqrt{m} \|W_{L+1} J_{\boldsymbol{\theta}_L}(\mathbf{x})\| \right\}$$

$$+ \sqrt{m} \gamma(\boldsymbol{\theta}_L, \tilde{\boldsymbol{\theta}}_L) \|W_{L+1}\| + \sqrt{m} \|\widetilde{W}_{L+1} - W_{L+1}\| \prod_{\ell=1}^{L} \|\widetilde{W}_\ell\|$$

$$= \sup_{t \in \mathcal{T}} V_t + \sqrt{m} \gamma(\boldsymbol{\theta}_L, \tilde{\boldsymbol{\theta}}_L) \|W_{L+1}\| + \sqrt{m} \|\widetilde{W}_{L+1} - W_{L+1}\| \prod_{\ell=1}^{L} \|\widetilde{W}_\ell\|,$$

where the final equality follows by recalling Definition B.1. Finally, the proof is concluded by noting that Proposition 3.7 implies $\mathrm{Lip}(f_{\tilde{\boldsymbol{\theta}}_{L+1}}, \mathcal{X}) \leq \sup_{\mathbf{x} \in \overline{\mathcal{B}}_{d_{\mathrm{in}}}(\mathbf{0},\Lambda)} \|J_{\tilde{\boldsymbol{\theta}}_{L+1}}(\mathbf{x})\|$. $\square$

The upper bound in Lemma B.6 consists of three components: the first controls the Lipschitz constant at initialization, while the second and third capture the effect of weight perturbations from initialization. Since Theorem 4.1 already provides a probabilistic bound for the first term, the following lemmas will focus on bounding the second and third terms in Equation (31).

**Lemma B.7.** *Let $W_1, \ldots, W_{L+1}$ be weight matrices, and let $R > 0$. Then, the following implication holds*

$$(W_1, \ldots, W_L) \in \mathcal{W}_{\mathrm{op}} \implies \sup_{\tilde{\boldsymbol{\theta}}_L \in \overline{\mathcal{B}}\left(\boldsymbol{\theta}_L, \frac{R}{\sqrt{m}}\right)} \gamma(\boldsymbol{\theta}_L, \tilde{\boldsymbol{\theta}}_L) \leq B_\gamma,$$

*where*

$$B_\gamma := L \cdot 2^L \left[ 2^{L+1} \beta_\phi \Lambda \left( \frac{R}{\sqrt{m}} \right)^{2L+1} \max\left\{ 1, e^2 \cdot \left( \frac{2\sqrt{m}}{R} \right)^{2L} \right\} + \left( \frac{R}{\sqrt{m}} \right)^{L+1} \max\left\{ 1, e \cdot \left( \frac{2\sqrt{m}}{R} \right)^{L} \right\} \right].$$

*Proof.* To simplify notation, we will denote $R' := \frac{R}{\sqrt{m}}$. We start by bounding the following term

$$
\sum_{k=1}^{L-r+1} \prod_{p=1}^{k+r-1} \|\widetilde{W}_p\| \leq \sum_{k=1}^{L-r+1} \prod_{p=1}^{k+r-1} (R' + \|W_p\|)
$$

$$
\overset{(\mathcal{W}_{\mathrm{op}})}{\leq} \sum_{k=1}^{L-r+1} \left( R' + 2 + \frac{2}{L} \right)^{k+r-1}
$$

$$
= \left( R' + 2 + \frac{2}{L} \right)^L \sum_{k=0}^{L-r} \left( R' + 2 + \frac{2}{L} \right)^{-k}
$$

$$
\overset{(a)}{\leq} \left( R' + 2 + \frac{2}{L} \right)^L \frac{R' + 2 + \frac{2}{L}}{R' + 1 + \frac{2}{L}}
$$

$$
\overset{(b)}{\leq} 2 \left( R' + 2 + \frac{2}{L} \right)^L
$$

$$
\overset{(c)}{\leq} 2^{L+1} \left( \frac{R}{\sqrt{m}} \right)^L \max \left\{ 1, e \cdot \left( \frac{2\sqrt{m}}{R} \right)^L \right\}, \tag{33}
$$

where (a) follows from the geometric series bound, (b) holds since $R' > 0$, and (c) follows from the inequality $x + y \leq 2\max\{x, y\}$ together with the substitution $R' = \frac{R}{\sqrt{m}}$. Moreover, we have that

$$
\sum_{r=1}^{L} \prod_{i=r+1}^{L} \|W_i\| \prod_{j=1}^{r-1} \|\widetilde{W}_j\| \overset{(\mathcal{W}_{\mathrm{op}})}{\leq} \sum_{r=1}^{L} \left( 2 + \frac{2}{L} \right)^{L-r} \left( R' + 2 + \frac{2}{L} \right)^{r-1}
$$

$$
\overset{(a)}{\leq} \sum_{r=1}^{L} \left( R' + 2 + \frac{2}{L} \right)^{L-1}
$$

$$
\overset{(b)}{\leq} L \cdot 2^L \left( \frac{R}{\sqrt{m}} \right)^L \max \left\{ 1, e \cdot \left( \frac{2\sqrt{m}}{R} \right)^L \right\} \tag{34}
$$

where (a) follows since $R' > 0$, and (b) from the basic inequalities $x + y \leq 2\max\{x, y\}$, and $(2 + \frac{2}{L})^L \leq e\, 2^L$. Hence, by recalling the definition of $\gamma(\boldsymbol{\theta}_L, \tilde{\boldsymbol{\theta}}_L)$, and substituting the bounds of Equation (33) and (34), we get

$$
\gamma(\boldsymbol{\theta}_L, \tilde{\boldsymbol{\theta}}_L) \leq \left[ 2^{L+1} \beta_\phi \Lambda \left( \frac{R}{\sqrt{m}} \right)^L \max \left\{ 1, e \cdot \left( \frac{2\sqrt{m}}{R} \right)^L \right\} + 1 \right] \sum_{r=1}^{L} \left( \prod_{i=r+1}^{L} \|W_i\| \right) \left( \prod_{j=1}^{r-1} \|\widetilde{W}_j\| \right) \|\widetilde{W}_r - W_r\|
$$

$$
\leq 2^L\, L \left[ 2^{L+1} \beta_\phi \Lambda \left( \frac{R}{\sqrt{m}} \right)^{2L} \max \left\{ 1, e^2 \cdot \left( \frac{2\sqrt{m}}{R} \right)^{2L} \right\} + \left( \frac{R}{\sqrt{m}} \right)^L \max \left\{ 1, e \cdot \left( \frac{2\sqrt{m}}{R} \right)^L \right\} \right] \frac{R}{\sqrt{m}} = B_\gamma.
$$

The result follows since the right-hand side of the above inequality does not depend on $\tilde{\boldsymbol{\theta}}_L$ and hence

$$
\sup_{\tilde{\boldsymbol{\theta}}_L \in \overline{\mathcal{B}}\left(\boldsymbol{\theta}_L, \frac{R}{\sqrt{m}}\right)} \gamma(\boldsymbol{\theta}_L, \tilde{\boldsymbol{\theta}}_L) \leq B_\gamma.
$$

$\square$

**Lemma B.8.** *Let $W_1, \ldots, W_{L+1}$ be weight matrices and let $R > 0$. Then, the following implication holds*

$$(W_1, \ldots, W_L) \in \mathcal{W}_{\mathrm{op}} \implies \sup_{\tilde{\boldsymbol{\theta}}_{L+1} \in \overline{\mathcal{B}}\left(\boldsymbol{\theta}_{L+1}, \frac{R}{\sqrt{m}}\right)} \left\{ \|\widetilde{W}_{L+1} - W_{L+1}\| \prod_{\ell=1}^{L} \|\widetilde{W}_\ell\| \right\}$$

$$\leq 2^L \left(\frac{R}{\sqrt{m}}\right)^{L+1} \max\left\{ 1, \, e \cdot \left(\frac{2\sqrt{m}}{R}\right)^L \right\}.$$

*Proof.* The result follows immediately with the following chain of inequalities

$$\sup_{\tilde{\boldsymbol{\theta}}_{L+1} \in \overline{\mathcal{B}}\left(\boldsymbol{\theta}_{L+1}, \frac{R}{\sqrt{m}}\right)} \left\{ \|\widetilde{W}_{L+1} - W_{L+1}\| \prod_{\ell=1}^{L} \|\widetilde{W}_\ell\| \right\} \leq \frac{R}{\sqrt{m}} \prod_{\ell=1}^{L} \left(\frac{R}{\sqrt{m}} + \|W_\ell\|\right) \leq \frac{R}{\sqrt{m}} \left(\frac{R}{\sqrt{m}} + 2 + \frac{2}{L}\right)^L$$

$$\leq 2^L \left(\frac{R}{\sqrt{m}}\right)^{L+1} \max\left\{ 1, \, e \cdot \left(\frac{2\sqrt{m}}{R}\right)^L \right\}$$

$\square$

Since we have now established bounds for all three terms appearing in Equation (31), we are prepared to prove the uniform Lipschitz perturbation bound stated in Theorem 4.2.

## B.6. Proof of Theorem 4.2

The proof is similar to that of Theorem 4.1. We start by defining the sets

$$\mathcal{E}_1 := \left\{ (W_1, \ldots, W_L) \, : \, \sup_{\tilde{\boldsymbol{\theta}}_L \in \overline{\mathcal{B}}\left(\boldsymbol{\theta}_L, \frac{R}{\sqrt{m}}\right)} \gamma(\boldsymbol{\theta}_L, \tilde{\boldsymbol{\theta}}_L) \leq B_\gamma \right\} \cap \mathcal{A}_1,$$

$$\mathcal{E}_2(W_1, \ldots, W_L) := \left\{ W_{L+1} \, : \, \|W_{L+1}\| \leq 3 \right\} \cap \mathcal{A}_2(W_1, \ldots, W_L),$$

$$\mathcal{E} := \left\{ (W_1, \ldots, W_{L+1}) \, : \, \sup_{h \in \mathcal{F}(\boldsymbol{\theta}_{L+1}, R)} \mathrm{Lip}(h, \mathcal{X}) \leq K_{\mathrm{init}} + C_{\mathrm{per}} \cdot R \right\}$$

By Lemmas B.5 and B.7, we know that $\mathcal{W}_{\mathrm{op}} \cap \mathcal{W}_{2 \to \infty} \subset \mathcal{E}_1$, and hence, $\mathbb{P}(\mathcal{E}_1) \geq \mathbb{P}(\mathcal{W}_{\mathrm{op}} \cap \mathcal{W}_{2 \to \infty})$. Furthermore, by Lemma B.4 we get $\mathbb{P}(\mathcal{E}_1) \geq \left(1 - 4\exp(-cm/L^2)\right)^L$, where $c > 0$ is an absolute constant. To bound the probability of $\mathcal{E}_2(W_1, \ldots, W_L)$, we have the following estimate

$$\mathbb{P}\left(\mathcal{E}_2(W_1, \ldots, W_L)\right) = 1 - \mathbb{P}\left(\mathcal{E}^c(W_1, \ldots, W_L)\right) \geq 1 - \mathbb{P}\left(\|W_{L+1}\| > 3\right) - \mathbb{P}\left(\mathcal{A}_2^c(W_1, \ldots, W_L)\right),$$

where we have used the union bound. By Lemma B.3 we know that $\mathbb{P}\left(\mathcal{A}_2^c(W_1, \ldots, W_L)\right) \leq 2\exp(-u^2)$ over the randomness of $W_{L+1}$. To bound the spectral norm of the final weight matrix $W_{L+1}$, we have that for any $s \geq 0$

$$\|W_{L+1}\| \leq \frac{1}{\sqrt{m}} \left(\sqrt{m} + \sqrt{d_{\mathrm{out}}} + s\right) \leq 3,$$

with probability at least $1 - 2\exp(-cm)$, which follows from Theorem 4.4.3 in (Vershynin, 2018) by setting $s = \sqrt{m}$ and recalling that by assumption the width satisfies $m \geq d_{\mathrm{out}}$. Hence, $\mathbb{P}(\|W_{L+1}\| > 3) \leq 2\exp(-cm)$, which yields

$$\mathbb{P}\left(\mathcal{E}_2(W_1, \ldots, W_{L+1})\right) \geq 1 - 2\exp(-cm) - 2\exp(-u^2).$$

Using Lemma B.6, it is easy to verify that if $(W_1, \ldots, W_L) \in \mathcal{E}_1$ and $W_{L+1} \in \mathcal{E}_2(W_1, \ldots, W_{L+1})$, then $(W_1, \ldots, W_{l+1}) \in \mathcal{E}$. Hence, Proposition C.1 in (Geuchen et al., 2025) implies that

$$\mathbb{P}(\mathcal{E}) \geq \mathbb{P}(\mathcal{E}_1) \inf_{(W_1, \ldots, W_L) \in \mathcal{E}_1} \mathbb{P}(\mathcal{E}_2(W_1, \ldots, W_L)) \geq \left(1 - 4\exp(-cm/L^2)\right)^L \left(1 - 2\exp(-cm) - 2\exp(-u^2)\right),$$

which is valid due to the independence of $W_{L+1}$ and $(W_1, \ldots, W_L)$. This concludes the proof.

## C. Perturbation Analysis

In this section, we present a set of four auxiliary perturbation inequalities—both with respect to the network's input and its weights—leveraging the smoothness and 1-Lipschitz property of the activation function.

The first lemma establishes a bound on the $\ell_2$-norm of the difference between the perturbed and unperturbed outputs of each intermediate layer. This bound is expressed in terms of the spectral norm of the perturbation, quantifying how much the perturbation affects the layer outputs.

**Lemma C.1.** *Let $W_1, \ldots W_{L+1}$ and $\widetilde{W}_1, \ldots \widetilde{W}_{L+1}$ be two collections of weight matrices. Then, if the activation function $\phi(\cdot)$ satisfies $\phi(0) = 0$ and is 1-Lipschitz, it holds that*

$$\left\| \tilde{\mathbf{x}}_\ell - \mathbf{x}_\ell \right\|_2 \ \leq \ \Lambda \sum_{r=1}^{\ell} \left( \prod_{j=1}^{r-1} \|W_j\| \right) \left( \prod_{i=r+1}^{\ell} \|\widetilde{W}_i\| \right) \|\widetilde{W}_r - W_r\|. \tag{35}$$

*for all $\ell \in [L]$ and $\mathbf{x} \in \overline{\mathcal{B}}_{d_{\mathrm{in}}}(\mathbf{0}, \Lambda)$.*

*Proof.* We will prove this inequality by induction over the layers. The induction hypothesis states that Eq. (35) holds for all $\ell \in [L]$. For the base case $\ell = 1$, it is obviously true since

$$\|\tilde{\mathbf{x}}_1 - \mathbf{x}_1\|_2 = \|\phi(\widetilde{W}_1 \mathbf{x}) - \phi(W_1 \mathbf{x})\|_2$$

$$\leq \|\widetilde{W}_1 \mathbf{x} - W_1 \mathbf{x}\|_2$$

$$\leq \Lambda \|\widetilde{W}_1 - W_1\|,$$

where the first inequality follows from the 1-Lipschitz property of the activation function $\phi(\cdot)$, and the second from the boundedness of the domain. Assuming that the hypothesis holds for some fixed $\ell \in \{1, 2, \ldots, L-1\}$, we have that

$$\|\tilde{\mathbf{x}}_{\ell+1} - \mathbf{x}_{\ell+1}\|_2 = \|\phi(\widetilde{W}_{\ell+1} \tilde{\mathbf{x}}_\ell) - \phi(W_{\ell+1} \mathbf{x}_\ell)\|_2$$

$$\leq \|\widetilde{W}_{\ell+1} \tilde{\mathbf{x}}_\ell - W_{\ell+1} \mathbf{x}_\ell\|_2$$

$$\leq \|\widetilde{W}_{\ell+1} \tilde{\mathbf{x}}_\ell - \widetilde{W}_{\ell+1} \mathbf{x}_\ell\|_2 + \|\widetilde{W}_{\ell+1} \mathbf{x}_\ell - W_{\ell+1} \mathbf{x}_\ell\|_2$$

$$\leq \|\widetilde{W}_{\ell+1}\| \|\tilde{\mathbf{x}}_\ell - \mathbf{x}_\ell\|_2 + \|\widetilde{W}_{\ell+1} - W_{\ell+1}\| \|\mathbf{x}_\ell\|_2.$$

By assumption, we can bound $\|\tilde{\mathbf{x}}_\ell - \mathbf{x}_\ell\|_2$ using Eq. (35). Moreover, noting that $\|\mathbf{x}_\ell\|_2 \leq \Lambda \prod_{j=1}^{\ell} \|W_j\|$ implies

$$\|\tilde{\mathbf{x}}_{\ell+1} - \mathbf{x}_{\ell+1}\|_2 \leq \Lambda \|\widetilde{W}_{\ell+1}\| \sum_{r=1}^{\ell} \left( \prod_{j=1}^{r-1} \|W_j\| \right) \left( \prod_{i=r+1}^{\ell} \|\widetilde{W}_i\| \right) \|\widetilde{W}_r - W_r\| + \Lambda \left( \prod_{j=1}^{\ell} \|W_j\| \right) \|\widetilde{W}_{\ell+1} - W_{\ell+1}\|$$

$$\leq \Lambda \sum_{r=1}^{\ell+1} \left( \prod_{j=1}^{r-1} \|W_j\| \right) \left( \prod_{i=r+1}^{\ell+1} \|\widetilde{W}_i\| \right) \|\widetilde{W}_r - W_r\|,$$

which proves the inductive step and the proof is concluded. □

The second perturbation inequality bounds the spectral norm of the difference between the network's Jacobian matrices at the penultimate layer, evaluated at two distinct points, in terms of the Euclidean distance between those points.

**Lemma C.2.** *Let $W_1, \ldots, W_{L+1}$ be weight matrices and $\phi(\cdot)$ a 1-Lipschitz and $\beta_\phi$-smooth activation function. Then, for all $\mathbf{x}, \mathbf{y} \in \mathbb{R}^{d_{\mathrm{in}}}$, the network's Jacobian matrix at the penultimate layer satisfies the following inequality*

$$\|J_{\boldsymbol{\theta}_L}(\mathbf{x}) - J_{\boldsymbol{\theta}_L}(\mathbf{y})\| \leq \beta_\phi \prod_{i=1}^{L} \|W_i\| \sum_{r=1}^{L} \|W_r\|_{2 \to \infty} \prod_{j=1}^{r-1} \|W_j\| \|\mathbf{x} - \mathbf{y}\|_2. \tag{36}$$

*Proof.* We will prove this result by induction. The induction hypothesis states that for every $\ell \in [L]$, and $\mathbf{x}, \mathbf{y} \in \mathbb{R}^{d_{\text{in}}}$,

$$\|J_{\boldsymbol{\theta}_\ell}(\mathbf{x}) - J_{\boldsymbol{\theta}_\ell}(\mathbf{y})\| \leq \beta_\phi \prod_{i=1}^{\ell} \|W_i\| \sum_{r=1}^{\ell} \|W_r\|_{2\to\infty} \prod_{j=1}^{r-1} \|W_j\| \|\mathbf{x} - \mathbf{y}\|_2 \tag{37}$$

For the base case $\ell = 1$, it holds true since

$$\|J_{\boldsymbol{\theta}_1}(\mathbf{x}) - J_{\boldsymbol{\theta}_1}(\mathbf{y})\| = \|D_{\boldsymbol{\theta}_1}(\mathbf{x})W_1 - D_{\boldsymbol{\theta}_1}(\mathbf{y})W_1\|$$
$$\leq \|D_{\boldsymbol{\theta}_1}(\mathbf{x}) - D_{\boldsymbol{\theta}_1}(\mathbf{y})\| \|W_1\|$$
$$= \|\phi'(W_1\mathbf{x}) - \phi'(W_1\mathbf{y})\|_\infty \|W_1\|$$
$$\leq \beta_\phi \|W_1(\mathbf{x} - \mathbf{y})\|_\infty \|W_1\|$$
$$\leq \beta_\phi \|W_1\| \|W_1\|_{2\to\infty} \|\mathbf{x} - \mathbf{y}\|_2,$$

where we have used the smoothness of the activation function and the fact that the spectral norm of a diagonal matrix equals the infinity norm of its entries. Assuming that the induction hypothesis holds for some fixed $\ell \in \{1, 2, \ldots, L-1\}$, for $\ell + 1$ we have that

$$\|J_{\boldsymbol{\theta}_{\ell+1}}(\mathbf{x}) - J_{\boldsymbol{\theta}_{\ell+1}}(\mathbf{y})\| = \|D_{\boldsymbol{\theta}_{\ell+1}}(\mathbf{x})W_{\ell+1}J_{\boldsymbol{\theta}_\ell}(\mathbf{x}) - D_{\boldsymbol{\theta}_{\ell+1}}(\mathbf{y})W_{\ell+1}J_{\boldsymbol{\theta}_\ell}(\mathbf{y})\|$$
$$\leq \|D_{\boldsymbol{\theta}_{\ell+1}}(\mathbf{x})W_{\ell+1}J_{\boldsymbol{\theta}_\ell}(\mathbf{x}) - D_{\boldsymbol{\theta}_{\ell+1}}(\mathbf{y})W_{\ell+1}J_{\boldsymbol{\theta}_\ell}(\mathbf{x})\|$$
$$+ \|D_{\boldsymbol{\theta}_{\ell+1}}(\mathbf{y})W_{\ell+1}J_{\boldsymbol{\theta}_\ell}(\mathbf{x}) - D_{\boldsymbol{\theta}_{\ell+1}}(\mathbf{y})W_{\ell+1}J_{\boldsymbol{\theta}_\ell}(\mathbf{y})\|$$
$$\leq \|D_{\boldsymbol{\theta}_{\ell+1}}(\mathbf{x}) - D_{\boldsymbol{\theta}_{\ell+1}}(\mathbf{y})\| \|W_{\ell+1}\| \|J_{\boldsymbol{\theta}_\ell}(\mathbf{x})\|$$
$$+ \|D_{\boldsymbol{\theta}_{\ell+1}}(\mathbf{y})\| \|W_{\ell+1}\| \|J_{\boldsymbol{\theta}_\ell}(\mathbf{x}) - J_{\boldsymbol{\theta}_\ell}(\mathbf{y})\|.$$

We proceed by bounding the individual terms of the above expression. By the smoothness of the activation function we derive

$$\|D_{\boldsymbol{\theta}_{\ell+1}}(\mathbf{x}) - D_{\boldsymbol{\theta}_{\ell+1}}(\mathbf{y})\| \leq \beta_\phi \|W_{\ell+1}\|_{2\to\infty} \|\mathbf{x}_\ell - \mathbf{y}_\ell\|_2$$

$$\leq \beta_\phi \|W_{\ell+1}\|_{2\to\infty} \prod_{i=1}^{\ell} \|W_i\| \|\mathbf{x} - \mathbf{y}\|_2$$

Moreover, the 1-Lipschitz property implies that $\|D_{\boldsymbol{\theta}_{\ell+1}}(\mathbf{x})\| \leq 1$ and $\|J_{\boldsymbol{\theta}_\ell}(\mathbf{x})\| \leq \prod_{j=1}^{\ell} \|W_j\|$. Finally, using the induction hypothesis (Eq. (37)) to bound $\|J_{\boldsymbol{\theta}_\ell}(\mathbf{x}) - J_{\boldsymbol{\theta}_\ell}(\mathbf{y})\|$, it follows that

$$\|J_{\boldsymbol{\theta}_{\ell+1}}(\mathbf{x}) - J_{\boldsymbol{\theta}_{\ell+1}}(\mathbf{y})\| \leq \beta_\phi \prod_{i=1}^{\ell+1} \|W_i\| \|W_{\ell+1}\|_{2\to\infty} \prod_{j=1}^{\ell} \|W_j\| \|\mathbf{x} - \mathbf{y}\|_2$$

$$+ \beta_\phi \prod_{i=1}^{\ell+1} \|W_i\| \sum_{r=1}^{\ell} \|W_r\|_{2\to\infty} \prod_{j=1}^{r-1} \|W_j\| \|\mathbf{x} - \mathbf{y}\|_2$$

$$= \beta_\phi \prod_{i=1}^{\ell+1} \|W_i\| \sum_{r=1}^{\ell+1} \|W_r\|_{2\to\infty} \prod_{j=1}^{r-1} \|W_j\| \|\mathbf{x} - \mathbf{y}\|_2.$$

Hence, the inductive step holds and the hypothesis is proven. $\square$

The next lemma also bounds the spectral norm of the Jacobian difference at the penultimate layer. However, unlike Lemma C.2, it considers perturbations with respect to the network's weights rather than its inputs.

**Lemma C.3.** *Let $\widetilde{W}_1, \ldots, \widetilde{W}_{L+1}$ and $W_1, \ldots, W_{L+1}$ be two collections of weight matrices. If the activation function $\phi(\cdot)$ satisfies the conditions of Assumption 3.4, then the Jacobian matrix at the penultimate layer satisfies the following inequality*

$$\|J_{\tilde{\boldsymbol{\theta}}_L}(\mathbf{x}) - J_{\boldsymbol{\theta}_L}(\mathbf{x})\| \leq \sum_{r=1}^{L} \left( \prod_{i=r+1}^{L} \|W_i\| \right) \left( \beta_\phi \Lambda \sum_{k=1}^{L-r+1} \prod_{p=1}^{k+r-1} \|\widetilde{W}_p\| + 1 \right) \left( \prod_{j=1}^{r-1} \|\widetilde{W}_j\| \right) \|\widetilde{W}_r - W_r\|, \qquad (38)$$

*for all $\mathbf{x} \in \overline{\mathcal{B}}_{d_{\text{in}}}(\mathbf{0}, \Lambda)$.*

*Proof.* We will prove this result by induction. The induction hypothesis states that for every $\ell \in [L]$ the following inequality holds

$$\|J_{\tilde{\boldsymbol{\theta}}_\ell}(\mathbf{x}) - J_{\boldsymbol{\theta}_\ell}(\mathbf{x})\| \leq \sum_{r=1}^{\ell} \left( \prod_{i=r+1}^{\ell} \|W_i\| \right) \left( \beta_\phi \Lambda \sum_{k=1}^{\ell-r+1} \prod_{p=1}^{k+r-1} \|\widetilde{W}_p\| + 1 \right) \left( \prod_{j=1}^{r-1} \|\widetilde{W}_j\| \right) \|\widetilde{W}_r - W_r\|. \qquad (39)$$

For the base case $\ell = 1$, we have that

$$\|J_{\tilde{\boldsymbol{\theta}}_1}(\mathbf{x}) - J_{\boldsymbol{\theta}_1}(\mathbf{x})\| = \left\| D_{\tilde{\boldsymbol{\theta}}_1}(\mathbf{x})\widetilde{W}_1 - D_{\boldsymbol{\theta}_1}(\mathbf{x})W_1 \right\|$$

$$\leq \|D_{\tilde{\boldsymbol{\theta}}_1}(\mathbf{x})\widetilde{W}_1 - D_{\boldsymbol{\theta}_1}(\mathbf{x})\widetilde{W}_1\| + \|D_{\boldsymbol{\theta}_1}(\mathbf{x})\widetilde{W}_1 - D_{\boldsymbol{\theta}_1}(\mathbf{x})W_1\|$$

$$\leq \|D_{\tilde{\boldsymbol{\theta}}_1}(\mathbf{x}) - D_{\boldsymbol{\theta}_1}(\mathbf{x})\| \|\widetilde{W}_1\| + \|\widetilde{W}_1 - W_1\|$$

$$\leq (\beta_\phi \Lambda \|\widetilde{W}_1\| + 1)\|\widetilde{W}_1 - W_1\|,$$

which is in agreement with the condition we wish to prove. The first inequality is an application of the triangle inequality, the second follows from the submultiplicativity of the spectral norm, whereas the third follows from the smoothness of the activation function and the relation $\|\widetilde{W}_1 - W_1\|_{2\to\infty} \leq \|\widetilde{W}_1 - W_1\|$. Assuming that the induction hypothesis holds for some fixed $\ell \in \{1, 2, \ldots, L-1\}$, at $\ell + 1$ we get

$$\|J_{\tilde{\boldsymbol{\theta}}_{\ell+1}}(\mathbf{x}) - J_{\boldsymbol{\theta}_{\ell+1}}(\mathbf{x})\| = \|D_{\tilde{\boldsymbol{\theta}}_{\ell+1}}(\mathbf{x})\widetilde{W}_{\ell+1} J_{\tilde{\boldsymbol{\theta}}_\ell}(\mathbf{x}) - D_{\boldsymbol{\theta}_{\ell+1}}(\mathbf{x})W_{\ell+1} J_{\boldsymbol{\theta}_\ell}(\mathbf{x})\|$$

$$\leq \underbrace{\|\text{diag}\big(\phi'(\widetilde{W}_{\ell+1}\tilde{\mathbf{x}}_\ell)\big)\widetilde{W}_{\ell+1} J_{\tilde{\boldsymbol{\theta}}_\ell}(\mathbf{x}) - \text{diag}\big(\phi'(W_{\ell+1}\tilde{\mathbf{x}}_\ell)\big)\widetilde{W}_{\ell+1} J_{\tilde{\boldsymbol{\theta}}_\ell}(\mathbf{x})\|}_{=:M_1}$$

$$+ \underbrace{\|\text{diag}\big(\phi'(W_{\ell+1}\tilde{\mathbf{x}}_\ell)\big)\widetilde{W}_{\ell+1} J_{\tilde{\boldsymbol{\theta}}_\ell}(\mathbf{x}) - \text{diag}\big(\phi'(W_{\ell+1}\mathbf{x}_\ell)\big)\widetilde{W}_{\ell+1} J_{\tilde{\boldsymbol{\theta}}_\ell}(\mathbf{x})\|}_{=:M_2} \qquad (40)$$

$$+ \underbrace{\|\text{diag}\big(\phi'(W_{\ell+1}\mathbf{x}_\ell)\big)\widetilde{W}_{\ell+1} J_{\tilde{\boldsymbol{\theta}}_\ell}(\mathbf{x}) - \text{diag}\big(\phi'(W_{\ell+1}\mathbf{x}_\ell)\big)W_{\ell+1} J_{\tilde{\boldsymbol{\theta}}_\ell}(\mathbf{x})\|}_{=:M_3}$$

$$+ \underbrace{\|\text{diag}\big(\phi'(W_{\ell+1}\mathbf{x}_\ell)\big)W_{\ell+1} J_{\tilde{\boldsymbol{\theta}}_\ell}(\mathbf{x}) - \text{diag}\big(\phi'(W_{\ell+1}\mathbf{x}_\ell)\big)W_{\ell+1} J_{\boldsymbol{\theta}_\ell}(\mathbf{x})\|}_{=:M_4}$$

We will proceed by bounding each one of the four terms appearing in the above decomposition separately.

**First term in Eq. (40):** Using the submultiplicativity of the spectral norm and the properties of the activation function, we

get

$$M_1 \leq \|\mathrm{diag}\big(\phi'(\widetilde{W}_{\ell+1}\tilde{\mathbf{x}}_\ell)\big) - \mathrm{diag}\big(\phi'(W_{\ell+1}\tilde{\mathbf{x}}_\ell)\big)\| \, \|\widetilde{W}_{\ell+1}\| \prod_{j=1}^{\ell}\|\widetilde{W}_j\|$$

$$\leq \beta_\phi \, \|\widetilde{W}_{\ell+1} - W_{\ell+1}\| \, \|\tilde{\mathbf{x}}_\ell\|_2 \, \|\widetilde{W}_{\ell+1}\| \prod_{j=1}^{\ell}\|\widetilde{W}_j\|$$

$$\overset{(a)}{\leq} \beta_\phi \Lambda \, \|\widetilde{W}_{\ell+1} - W_{\ell+1}\| \prod_{p=1}^{\ell+1}\|\widetilde{W}_p\| \prod_{j=1}^{\ell}\|\widetilde{W}_j\|.$$

**Second term in Eq.** (40)**:** Using Lemma C.1 (with the role of $\boldsymbol{\theta}_\ell$ and $\tilde{\boldsymbol{\theta}}_\ell$ interchanged) to bound the layer-wise perturbation $\|\tilde{\mathbf{x}}_\ell - \mathbf{x}_\ell\|_2$, we obtain

$$M_2 \leq \|\mathrm{diag}\big(\phi'(W_{\ell+1}\tilde{\mathbf{x}}_\ell)\big) - \mathrm{diag}\big(\phi'(W_{\ell+1}\mathbf{x}_\ell)\big)\| \prod_{p=1}^{\ell+1}\|\widetilde{W}_p\|$$

$$\leq \beta_\phi \|W_{\ell+1}\| \, \|\tilde{\mathbf{x}}_\ell - \mathbf{x}_\ell\|_2 \prod_{p=1}^{\ell+1}\|\tilde{W}_p\|$$

$$\overset{(b)}{\leq} \beta_\phi \Lambda \sum_{r=1}^{\ell} \left( \prod_{i=r+1}^{\ell+1}\|W_i\| \right) \left( \prod_{j=1}^{r-1}\|\widetilde{W}_j\| \right) \left( \prod_{p=1}^{\ell+1}\|\widetilde{W}_p\| \right) \|\widetilde{W}_r - W_r\|.$$

**Third term in Eq.** (40)**:** We have

$$M_3 \overset{(c)}{\leq} \|\widetilde{W}_{\ell+1} - W_{\ell+1}\| \prod_{j=1}^{\ell}\|\widetilde{W}_j\|.$$

**Fourth term in Eq.** (40)**:.** Using the induction hypothesis of Eq. (39) to bound the Jacobian perturbation error $\|J_{\tilde{\boldsymbol{\theta}}_\ell}(\mathbf{x}) - J_{\boldsymbol{\theta}_\ell}(\mathbf{x})\|$ at the previous layer, we get

$$M_4 \leq \|W_{\ell+1}\| \, \|J_{\tilde{\boldsymbol{\theta}}_\ell}(\mathbf{x}) - J_{\boldsymbol{\theta}_\ell}(\mathbf{x})\|$$

$$\leq \|W_{\ell+1}\| \sum_{r=1}^{\ell} \left( \prod_{i=r+1}^{\ell}\|W_i\| \right) \left( \beta_\phi \Lambda \sum_{k=1}^{\ell-r+1} \prod_{p=1}^{k+r-1}\|\widetilde{W}_p\| + 1 \right) \left( \prod_{j=1}^{r-1}\|\widetilde{W}_j\| \right) \|\widetilde{W}_r - W_r\|$$

$$\overset{(d)}{=} \sum_{r=1}^{\ell} \left( \prod_{i=r+1}^{\ell+1}\|W_i\| \right) \left( \beta_\phi \Lambda \sum_{k=1}^{l-r+1} \prod_{p=1}^{k+r-1}\|\widetilde{W}_p\| + 1 \right) \left( \prod_{j=1}^{r-1}\|\widetilde{W}_j\| \right) \|\widetilde{W}_r - W_r\|.$$

By grouping the bounds $(a)$ and $(c)$ established for the terms $M_1$ and $M_3$, we obtain

$$M_1 + M_3 \leq \left( \beta_\phi \Lambda \prod_{p=1}^{\ell+1}\|\widetilde{W}_p\| + 1 \right) \left( \prod_{j=1}^{\ell}\|\widetilde{W}_j\| \right) \|\widetilde{W}_{\ell+1} - W_{\ell+1}\| \tag{41}$$

Similarly, combining the bounds $(b)$ and $(d)$ established for the terms $M_2$ and $M_4$ implies

$$M_2 + M_4 \leq \sum_{r=1}^{\ell} \left( \prod_{i=r+1}^{\ell+1} \|W_i\| \right) \left( \beta_\phi \Lambda \sum_{k=1}^{\ell-r+1} \prod_{p=1}^{k+r-1} \|\widetilde{W}_p\| + \beta_\phi \Lambda \prod_{p=1}^{\ell+1} \|\widetilde{W}_p\| + 1 \right) \left( \prod_{j=1}^{r-1} \|\widetilde{W}_j\| \right) \|\widetilde{W}_r - W_r\|$$

$$= \sum_{r=1}^{\ell} \left( \prod_{i=r+1}^{\ell+1} \|W_i\| \right) \left( \beta_\phi \Lambda \sum_{k=1}^{\ell-r+2} \prod_{p=1}^{k+r-1} \|\widetilde{W}_p\| + 1 \right) \left( \prod_{j=1}^{r-1} \|\widetilde{W}_j\| \right) \|\widetilde{W}_r - W_r\|. \quad (42)$$

Finally, by combining Eq. (41) and (42), we conclude that

$$\|J_{\tilde{\boldsymbol{\theta}}_{\ell+1}}(\mathbf{x}) - J_{\boldsymbol{\theta}_{\ell+1}}(\mathbf{x})\| \leq (M_1 + M_3) + (M_2 + M_4)$$

$$\leq \left( \beta_\phi \Lambda \prod_{p=1}^{\ell+1} \|\widetilde{W}_p\| + 1 \right) \left( \prod_{j=1}^{\ell} \|\widetilde{W}_j\| \right) \|\widetilde{W}_{\ell+1} - W_{\ell+1}\|$$

$$+ \sum_{r=1}^{\ell} \left( \prod_{i=r+1}^{\ell+1} \|W_i\| \right) \left( \beta_\phi \Lambda \sum_{k=1}^{\ell-r+2} \prod_{p=1}^{k+r-1} \|\widetilde{W}_p\| + 1 \right) \left( \prod_{j=1}^{r-1} \|\widetilde{W}_j\| \right) \|\widetilde{W}_r - W_r\|$$

$$= \sum_{r=1}^{\ell+1} \left( \prod_{i=r+1}^{\ell+1} \|W_i\| \right) \left( \beta_\phi \Lambda \sum_{k=1}^{\ell-r+2} \prod_{p=1}^{k+r-1} \|\widetilde{W}_p\| + 1 \right) \left( \prod_{j=1}^{r-1} \|\widetilde{W}_j\| \right) \|\widetilde{W}_r - W_r\|,$$

which proves the inductive step, and thus the hypothesis holds for all $\ell \in [L]$. $\qquad \square$

## D. Technical Lemmas

**Lemma D.1.** *Let $A \in \mathbb{R}^{m \times d}$ be a random matrix with independent standard Gaussian entries $[A]_{i,j} \sim \mathcal{N}(0,1)$ for $(i,j) \in [m] \times [d]$. Then, for all $u \geq 0$, it holds that*

$$\|A\|_{2 \to \infty} \leq \sqrt{d} + C\sqrt{\log m} + u,$$

*with probability at least $1 - 2\exp\left(-cu^2\right)$, where $C, c > 0$ are absolute constants.*

*Proof.* We start by noting that

$$\|A\|_{2 \to \infty} = \sup_{\|\mathbf{x}\|_2 = 1} \|A\mathbf{x}\|_\infty = \max_{1 \leq i \leq m} \|\mathbf{a}^{(i)}\|_2,$$

where $\mathbf{a}^{(i)} \in \mathbb{R}^d$ denotes the $i$-th row of matrix $A$ for $i \in [m]$. Thus, the expected value of $\|A\|_{2 \to \infty}$ satisfies

$$\mathbb{E}\|A\|_{2 \to \infty} = \sqrt{d} + \mathbb{E}\left[ \max_{1 \leq i \leq m} \left( \|\mathbf{a}^{(i)}\|_2 - \sqrt{d} \right) \right].$$

Let $X_i := \|\mathbf{a}^{(i)}\|_2 - \sqrt{d}$ for $i \in [m]$. By Theorem 3.1.1 in (Vershynin, 2018), we know that $\|X_i\|_{\psi_2} = \mathcal{O}(1)$. Moreover, Proposition 2.7.6 in (Vershynin, 2018) implies that

$$\mathbb{E} \max_{1 \leq i \leq m} |X_i| \leq C\sqrt{\log m},$$

where $C > 0$ is an absolute constant. The mapping $A \mapsto \|A\|_{2 \to \infty}$ is 1-Lipschitz with respect to the Euclidean (Frobenius) metric, since for any $A_1, A_2 \in \mathbb{R}^{m \times d}$ we have

$$\left| \|A_1\|_{2 \to \infty} - \|A_2\|_{2 \to \infty} \right| \leq \|A_1 - A_2\|_{2 \to \infty} \leq \|A_1 - A_2\|_{\mathrm{F}}$$

Hence, by the concentration of Lipschitz functions (Theorem 5.2.3 in (Vershynin, 2018)) it follows that for any $u \geq 0$,

$$\|A\|_{2 \to \infty} \leq \mathbb{E}\|A\|_{2 \to \infty} + u \leq \sqrt{d} + C\sqrt{\log m} + u,$$

with probability at least $1 - 2\exp(-cu^2)$. $\qquad \square$

**Lemma D.2.** *Let* $(\mathcal{T}_1, \rho_1), \ldots (\mathcal{T}_n, \rho_n)$ *be metric spaces and* $\mathcal{S}_i \subset \mathcal{T}_i$ *for* $i \in [n]$, *where* $n \in \mathbb{N}$ *is some natural number. Then, for any* $\varepsilon > 0$ *and* $\alpha_1, \alpha_2, \ldots, \alpha_n > 0$, *the covering number of the cartesian product* $\mathcal{S} := \mathcal{S}_1 \times \cdots \times \mathcal{S}_n$ *satisfies*

$$N(\mathcal{S}, \rho, \varepsilon) \leq \prod_{i=1}^n N(\mathcal{S}_i, \rho_i, \frac{\varepsilon}{n\alpha_i}),$$

*with respect to the metric* $\rho : \mathcal{T} \times \mathcal{T} \to \mathbb{R}_{\geq 0}$ *given by*

$$\rho(x, y) := \sum_{i=1}^n \alpha_i \rho_i(x_i, y_i) \quad \text{for all } x, y \in \mathcal{T}, \tag{43}$$

*where* $\mathcal{T} := \mathcal{T}_1 \times \cdots \times \mathcal{T}_n$.

*Proof.* Let $\mathcal{G}_i$ be a minimum cardinality $\varepsilon/n\alpha_i$-net of $\mathcal{S}_i$ for $i \in [n]$. Then, there exists some $y_i \in \mathcal{G}_i$ for all $i \in [n]$ such that

$$\rho(x, y) = \sum_{i=1}^n \alpha_i \rho_i(x_i, y_i) \leq \sum_{i=1}^n \alpha_i \frac{\varepsilon}{n\alpha_i} = \varepsilon \quad \text{for all } x \in \mathcal{S}.$$

Hence, the set $\mathcal{G} := \mathcal{G}_1 \times \cdots \times \mathcal{G}_n$ is an $\varepsilon$-net of $\mathcal{S}$ with respect to the metric $\rho(\cdot)$, and we conclude that

$$N(\mathcal{S}, \rho, \varepsilon) \leq \prod_{i=1}^n |\mathcal{G}_i| = \prod_{i=1}^n N(\mathcal{S}_i, \rho_i, \frac{\varepsilon}{n\alpha_i}).$$

$\square$

