# OpenReview forum: "Width Independent Bounds for the Local Lipschitz Constant of Deep Neural Networks at Random Initialization and after Lazy Training"
_ICML.cc/2026/Conference — ICML 2026 regular_

### Official Review · Reviewer_7zWK · 2026-03-05

**Soundness:** 3
**Presentation:** 3
**Significance:** 2
**Originality:** 3
**Overall Recommendation:** 3
**Confidence:** 3

**Summary:**

This paper theoretically analyzes the Lipschitz constant of randomly initialized neural networks and those obtained from lazy training. The main objective of the authors is to provide an upper-bound of the Lipschitz constant that does not depend on the width size, a behavior suggested by numerous empirical works. They successfully this goal by employing analytical tools such as Dudley’s integral similar to prior work (Geuchen et al., 2025) and perturbation theory. The theoretical contribution is illustrated by some synthetic small-scale experiments.

**Compliance With Llm Reviewing Policy:**

Affirmed.

**Final Justification:**

After the rebuttal, I believe the technical soundness and presentation of this paper meets the standard of a scientific paper. They successfully eliminate the dependence on $m$, which I believe is a particularly interesting theoretical result. Moreover, the extension to the lazy regime is indeed novel and appears promising for future work. This reflects my score 3 for originality (which I increased from 2 after the rebuttal).

My main reason for initial rejection was the insufficient explicit comparison with prior work. The paper could be interpreted as eliminating the $m$ dependence without clearly stating that prior works already achieve a $\sqrt{\log m}$ dependence for ReLU activation functions, which is really slow in non-asymptotic regimes. The authors have suggested adding a more explicit comparison in the revised version and nuanced statements throughout the paper.

Now, for a theoretical paper, the proof strategy and, more importantly, the underlying reason why the dependence on $m$ can be removed is not discussed in sufficient depth. As a result, the paper does not fully deliver the level of theoretical insight one might expect. Notably, the authors use a proof strategy similar to that of Geuchen et al (2025) that already enables for ReLU activation function to obtain a $\sqrt{\log m}$ dependence. While I acknowledge that the authors consider a more general class of smooth activation fuctions, this leaves doubts about the theoretical novelty and impact of Theorem 4.1.

In my view, the motivation for improving the dependence from $\sqrt{\log m}$ remains also somewhat weak, and the rebuttal did not sufficiently address this point. In particular, I have the following concerns:

1) What are the concrete theoretical obstacles to retaining a $\sqrt{\log m}$ dependence?
2) What are the concrete practical drawbacks of retaining such a $\sqrt{\log m}$ dependence?

Therefore, I have increased my score to weak reject.

**Key Questions For Authors:**

Q1. Could the authors compare their problem setting and result with Dirksen et al. (2025) and Geuchen et al. (2025)? Please focus on the improvements achieved in the paper over those previous works.

Q2. How does the Lipschitz constant evolve with respect to the depth? An empirical investigation of the effect of depth similarly to Figure 1 may help to elucidate this.

Q3. Still related to Q2, can the exponential dependence on $L$ be improved? It seems the bounds of Dirksen et al. (2025) do not present such dependency, and using their problem setting may help to avoid this.

**Limitations:**

Yes

**Strengths And Weaknesses:**

(Strengths)

S1. The theoretical analysis seems to be correct. No apparent major issues in the proofs.

S2. This paper is well-written. Notably, every section is carefully introduced which helps the reader to follow the arguments of the paper.

S3. The extension of the Lipschitz bound to the domain attained by lazy training is interesting and may influence future research leading to new insights on the behavior of neural networks.

S4 The paper successfully erases the dependence of the Lipschitz bounds on the width, which aligns with empirical results. This idea is a relevant approach in understanding the essential behavior of neural networks.

(Weaknesses)

W1. The comparison with prior relevant results is limited. While the background is well-documented, the explicit comparison of the bounds provided in this work (e.g., Theorem 4.1) and those of previous papers which consider similar scenarios such as Dirksen et al. (2025) and Geuchen et al. (2025) is missing. To clearly assess the advance made in this paper and the validity of authors’ assumptions and problem setting, such a discussion is primordial to be included in the paper.

W2. While the motivation to eliminate the $m$ dependence of the bound is well-explained and constitutes one of the main contributions of this paper, this result seems to be somewhat incremental.  Geuchen et al. (2025) studied a closely related problem setting and derived an upper bound with logarithmic dependence on $m$ (same for Dirksen et al. (2025)). Since logarithmic dependence is usually considered as “acceptable” in the machine learning community, the improvement attained by the authors is relatively moderate.

W3. The bounds provided is exponentially dependent on $L$, the depth of the neural network. As a result, the obtained bound may quickly become an overestimation for deep neural networks. This issue is not present in the trivial bound (equation (9)) as $\|W_l\|=O(1)$ with high probability. Moreover, the experiments were conducted only on shallow networks, and it is unclear whether the conclusions scale to deeper architectures.

---

> ### Author Rebuttal · Authors · 2026-03-29
>
> We thank the reviewer for their constructive feedback and for the time and effort invested in evaluating our work. Below, we address the expressed concerns in detail.
>
> ## Weaknesses
>
> > W2. While the motivation to eliminate the $m$ dependence...
>
> The main contributions of this work are twofold. First, we establish width-independent bounds for the Lipschitz constant of neural networks at initialization, in the spirit of results by Geuchen et al. (2025) and Dirksen et al. (2025). Our approach differs in three key aspects:
> Firstly, we consider a broad class of smooth activation functions—reflecting their growing practical relevance.
> Secondly, as you mention, we eliminate the logarithmic dependence on network width that appears in prior bounds. Even though removing a logarithmic factor is a marginal improvement in many applications, we believe that here it is crucial:   such a factor prevents combining the robustness result with the NTK theory, which  studies the asymptotic behavior of neural networks in the infinite width limt and hence cannot accomodate even a logarithmic width dependence.
> Thirdly, we derive from our results post-training guarantees  for a class of networks operating in the lazy regime. To the best of our knowledge, such results have not been established in previous work.
>
> > W3. The bounds provided is exponentially dependent on...
>
> Indeed, the exponential growth is the main limitation of our work, a point we clearly stress in the Conclusions section of our paper. However, we argue that this behavior is also present in the trivial bound of Equation (9). If we consider the initialization scheme of Assumption 3.4, a standard result from random matrix theory indicates that for any $u \geq 0$,
>
> - For $\ell = 1$: $|| W_{1}|| \leq \frac{1}{\sqrt{m}}(\sqrt{d_{\mathrm{in}}} + \sqrt{m} + u) \leq 2 + \frac{u}{\sqrt{m}}$ with probability at least $1 - \exp(-c u^{2})$
> - For $\ell = 2, \dots, L$: $|| W_{\ell}|| \leq \frac{1}{\sqrt{m}}(\sqrt{m} + \sqrt{m} + u) \leq 2 + \frac{u}{\sqrt{m}}$ with probability at least $1 - \exp(-c u^{2})$
> - For $\ell = L+1$: $|| W_{L+1}|| \leq \frac{1}{\sqrt{m}}(\sqrt{m} + \sqrt{d_{\mathrm{out}}}+ u) \leq 2 + \frac{u}{\sqrt{m}}$ with probability at least $1 - \exp(-c u^{2})$
>
> By setting $u = \sqrt{m}$ and using the independence of the weight matrices across different layers, we conclude that
> $$\sqrt{m} \prod_{\ell=1}^{L+1} ||W_{\ell}|| \leq \sqrt{m} \cdot 3^{L+1},$$
> with probability at least $(1 - \exp(-cm))^{L+1}$. Thus, the exponential dependence on depth is already inherent in the trivial bound and is not introduced by our proof method.
>
> ## Key Questions For Authors
>
> > Q1. Could the authors compare their problem setting...
>
> Thank you for the proposal. We agree that a clear comparison between these closely related works will strengthen the presentation and clarity of our paper. In the camera-ready version of the paper we will include a table in the introduction to indicate the differences in the setting and results. A first draft of such a table can be seen below:
>
> | Method           | Activation | Bound                                                                 | Lazy Training                                                                 |
> |------------------|------------|-----------------------------------------------------------------------|--------------------------------------------------------------------------------|
> | **Ours**         | Smooth     | $\mathcal{O}\left(2^{\mathcal{O}(L)} \sqrt{d}\right)$              | $\mathcal{O}\left(2^{\mathcal{O}(L)} (\sqrt{d} +  L \cdot R)\right)$ |
> | Geuchen et al.   | ReLU       | $\mathcal{O}\left(2^{\mathcal{O}(L)} \sqrt{L\cdot d \cdot\log\left(\tfrac{m}{d}\right)}\right)$ | –                                                                              |
> | Dirksen et al.   | ReLU       | $\mathcal{O}\left(\sqrt{d \cdot\log\left(\tfrac{m}{d}\right)}\right)$ | –                                                                              |
>
> > Q2 How does the Lipschitz constant evolve with respect to...
>
> Thank you for the suggestion. In the camera-ready version of the paper we will include an additional experiment to examine how the Lipschitz constant scales with network depth. However, we do not expect to observe exponential growth in practice, particularly for smooth activation functions that approximate ReLU. As you correctly noted, such exponential dependence has been eliminated in the ReLU setting by Dirksen et al. (2025).
>
> > Q3  Still related to Q2, can the exponential dependence on...
>
> At present, it remains unclear whether the exponential dependence can be avoided. While the results of Dirksen et al. (2025) do not exhibit such a dependence, their analysis relies crucially on the structure of activation patterns in ReLU networks, which cannot be easily transferred to our setting. Moreover, their work focuses on a particular activation function (ReLU), whereas we consider a broader class of smooth activations.

---

> > ### Author Rebuttal · Reviewer_7zWK · 2026-04-02
> >
> > Thank you very much for the comprehensive response. My questions are mostly addressed, and I strongly believe that the inclusion of a comparative table is essential for the paper. In particular, the improvement should be stated explicitly, namely that your bound removes the $\sqrt{\log m}$ factor.  For example, in l.236 left column, the authors write “A central goal of this work is to eliminate the undesired $\sqrt{m}$ dependence.” This should be more nuanced from this perspective.
> >
> > Regarding the weaknesses, my concerns are partially addressed.
> > For W3, thank you for the clarification. I believe this discussion could be incorporated into the paper to emphasize that dependence on $L$ is ubiquitous in most existing bounds under your assumptions. While I understand that analyzing the dependence on $L$ is not the primary focus of this work, readers may otherwise misinterpret the results as being independent of $m$ at the expense of a potentially stronger dependence on $L$, which could make the bounds effectively looser.
> >
> > Concerning W2, I understand that the scope of the paper is *overparameterized shallow* networks, and therefore removing the logarithmic dependence is necessary to avoid vacuous bounds. While this is technically sound, I still have some doubts regarding the practical usefulness of this improvement. Could you provide more concrete theoretical motivations or practical scenarios in which vacuous Lipschitz bounds would be problematic or limiting?
> >
> > Given that this is primarily a theoretical paper, I would also like to ask the following questions. What is the technical novelty of specifically Theorem 4.1 compared to Geuchen et al. (2025) and Dirksen et al. (2025)? What were the main technical challenges in extending the result to smooth functions? What is the key effect or mechanism that allows the logarithmic term to be removed?
> >
> > I will update my score accordingly. Clarifying the above questions would enable a more thorough evaluation of the paper.

---

> > > ### Author Response · Authors · 2026-04-04
> > >
> > > Thank you very much for your constructive feedback.
> > >
> > > > Response to the first remark:
> > >
> > > Regarding your first comment on our phrasing in I.236: We have sought to clarify in both the introduction and the abstract that, for ReLU networks, existing results exhibit a $\sqrt{\log m}$ dependence on the width, rather than a $\sqrt{m}$ dependence. This improvement, however, does not currently extend to networks with smooth activation functions, which motivated our original phrasing. For greater clarity, in the camera-ready version we will revise the sentence to: “A central goal of this paper is to eliminate the undesirable $\sqrt{\log m}$ dependence and to extend the results beyond a single activation function to a broad class of smooth activations, for which only a $\sqrt{m}$ dependence was previously known.”
> > >
> > > > Response to the second remark:
> > >
> > > Regarding your second comment on the exponential dependence on depth, we agree that including a brief discussion of the behavior of the existing trivial bound would improve the clarity of the paper. In the camera-ready version, we will therefore add a short paragraph along the lines of our response to Remark W3.
> > >
> > > > Regarding your third remark:
> > >
> > > Our bounds capture a general robustness phenomenon that applies to both deep and shallow networks, provided they operate in the lazy/NTK regime. However, as you noted, their practical relevance diminishes as network depth increases. We therefore argue that their primary practical value lies in the setting of shallow and wide networks. In this regime, we demonstrate indicative width independence for a broad class of smooth activation functions, both at random initialization and, more importantly, after training. To the best of our knowledge, this is the first work to establish Lipschitz bounds for trained networks beyond the naive estimates for any regime.
> > >
> > > > Response to the fourth remark:
> > >
> > > Regarding your final question on the main technical novelty underlying the proof of Theorem 4.1, our approach follows a similar high-level strategy to that of Geuchen et al. (2025). In particular, we express the network's Lipschitz constant as the supremum of a stochastic process and subsequently apply Dudley’s integral. The key remaining step is to bound the metric entropy of the process’s index set in order to obtain a concrete estimate.
> > >
> > > The authors of Geuchen et al. (2025) rely on a well-known property of ReLU networks: the number of distinct activation patterns grows only polynomially with the network width, rather than exponentially. Exploiting this fact, they construct a covering that accounts for all such activation patterns. As a result, the covering number of the index set scales with the width, which in turn yields a logarithmic dependence on $m$.
> > >
> > > In contrast, this approach does not extend to our setting, since activation patterns are not well-defined for networks with smooth activation functions. Thus, from a technical perspective, this is not merely about removing a logarithmic factor, but rather a $\sqrt{m}$ factor that arises in this more general setting. We exploit smoothness to show in Lemma B.2 that the subgaussian norm of the process's increments is controlled by a metric $\rho(\cdot)$ defined on a set $T$ that is independent of the network's architecture and only depends on the diameter of the input set, the input dimension, and the output dimension. This enables us to bound the covering number of $T$ with respect to $\rho$ without introducing any dependence on the width.
> > >
> > > The main challenge in extending these results to networks with smooth activation functions lies in establishing the perturbation inequalities given in Lemmas C.1 and C.3, which relate perturbations of the weights to perturbations of the post-activations and of the network’s Jacobian matrix, respectively. In particular, Lemma C.1 is instrumental in proving Lemma B.2, which is a key ingredient in Theorem 4.1 (Lipschitz bound at initialization). Similarly, Lemma C.3 underpins Lemma B.6, which is essential for Theorem 4.2 (uniform Lipschitz bound in the lazy regime).

---

### Official Review · Reviewer_g2ws · 2026-03-06

**Soundness:** 3
**Presentation:** 3
**Significance:** 3
**Originality:** 3
**Overall Recommendation:** 5
**Confidence:** 3

**Summary:**

This papers studies the Lipschitz constant of neural networks with smooth activation functions, when the weights are randomly initialized, under a lazy training regime. Specifically, the authors handle jointly the cases of random initialization and lazy training, while prior works mainly focused on the Lipschitz constant of randonly initialized networks, without taking into account the lazy training regime.

The authors treat two cases: that of randomly initialized networks, and then examine randomly initialized networks that are lazely trained. In both cases, they give probabilistic upper bounds on the Lipschitz constants of the examined networks. These bounds are independent of the network width, under mild assumptions on the network width, the activation functions, the input data, and the weights. The main novelty in the proposed framework lies in explaining how the random iniitalization, along with the implicit regularization of lazy training, ripple out to the Lipschitz constant of fully connected networks, thereby shedding light into their generalization and adversarial robustness.

Finally, the validity of the presented theory is assessed through a series of experiments. Results conform with derived theory, and adequately explain the phenomenon observed in theory.

**Compliance With Llm Reviewing Policy:**

Affirmed.

**Final Justification:**

The authors addressed all my concerns, so I will keep my original score (Accept).

**Key Questions For Authors:**

1. Generally, it is advised that claims are supported by relevant theory and practice. This relates to my concern on the (adversarial) robustness of networks and how it is reflected in the Lipschitz constants. It's good practice that you mention the effect and importance of Lipschitz constants in the networks' robustness, but can you provide an experiment, even small one, regarding the efficiency of your derived method, when the networks under study are attacked, e.g., by PGD? Or at least stress how robustness could be implied from your experimental results?
2. Could you please provide some more details on the results of Sec. 6.2? Is the behavior observed in your experiments anticipated or not? How would you explain the results of Table 1? I think that details like these would better position the practicality of the paper.
3. Could you please amend Sec. 5, so that it gives higher priority to what is novel in your approach, rather than the technicality of the proofs itself? What is presently written in Sec. 5 could be postponed for a relevant Appendix Section.
4. Another rule-of-thumb is that all results proved in a paper, basically constitute its "Main Results". In the way that the paper is presently written, I feel that Proposition 3.6, which is a theoretical result of yours, remains "hidden" in the background Sec. 3. Could you reorganize things a bit, so that Proposition 3.6 is placed in the "Main Results" Sec. 4, where it belongs? The same holds for Definition 3.7.

**Limitations:**

Yes.

**Strengths And Weaknesses:**

## Strengths

The paper is well-written and thus easily readable. A comprehensive review of related work in all aspects covered throughout the paper demonstrates a clear understanding of what is missing from related work and how to address it. The mathematical parts seem solid and the results novel, and everything is supported with relevant claims/assumptions. Finally, the authors provide a reasonable amount of experimental results.

## Weaknesses

1. Although it is well-known - and justified within the paper - how Lipschitz constants relate to the (adversarial) robustness of models, I don't see any relevant results in the experimental section.
2. The variety of experimental results is satisfactory, yet, the discussion remains slightly "neglected". For instance, in Sec. 6.2, it feels like more details are given in the setup per se, rather than explaining what is the overall take-away message, as this could be depicted in the "Observations" paragraph.
3. Section 5 seems a bit odd: this is what one would normally see in an Appendix, not in the main paper. This deprives the paper of additional space, which could be leveraged, e.g., to address the previous two points a bit more.
4. Minor, but still: I think that Proposition 3.6 isn't placed correctly in text (see suggestions in the key questions).

---

> ### Author Rebuttal · Authors · 2026-03-29
>
> We thank the reviewer for their constructive feedback and for the time and effort invested in evaluating our work. Below, we address the expressed concerns in detail.
>
> ## Key Questions For Authors
>
> > 1. Generally, it is advised that claims are supported by relevant theory and practice. This relates to my concern on the (adversarial) robustness of networks and how it is reflected in the Lipschitz constants. It's good practice that you mention the effect and importance of Lipschitz constants in the networks' robustness, but can you provide an experiment, even small one, regarding the efficiency of your derived method, when the networks under study are attacked, e.g., by PGD? Or at least stress how robustness could be implied from your experimental results?
>
> Relating the Lipschitz constant to the robustness properties of neural networks serves as the primary motivation for establishing our width-independent bounds. In the camera-ready version of the paper, we will further elaborate on relevant experimental findings in the literature (see, e.g., Robustness in Deep Learning: The Good (Width), the Bad (Depth), and the Ugly (Initialization)), which highlight the relationship between network width and robustness. However, due to space and time constraints, we are unable to include a detailed empirical evaluation in the current version.
>
> > 2. Could you please provide some more details on the results of Sec. 6.2? Is the behavior observed in your experiments anticipated or not? How would you explain the results of Table 1? I think that details like these would better position the practicality of the paper.
>
> The results align with theoretical expectations. In particular, Figure 2 indicates that the student network operates in the lazy regime during training: the relative distance between the trained weights and their initialization decreases with increasing width across all layers, and the observed $m^{-1/2}$ scaling confirms that the model lies in the lazy training class $\mathcal{F}(\mathbf{\theta_{0}}, R)$. Moreover, Table 1 empirically supports Theorem 4.2: regardless of width, the student network achieves training error below the noise level (i.e., training with SGD leads to overfitting), while the empirical Lipschitz constant remains approximately unchanged.
>
> The detailed description of the experimental setup in Section 6.2 is intended to ensure reproducibility. However, we agree that greater emphasis should be placed on interpreting the results rather than on experimental details. In the camera-ready version of the paper, we will revise this section to further elaborate on the experimental observations and their implications.
>
> > 3. Could you please amend Sec. 5, so that it gives higher priority to what is novel in your approach, rather than the technicality of the proofs itself? What is presently written in Sec. 5 could be postponed for a relevant Appendix Section.
>
> Thank you for the suggestion. As this is primarily a theoretical study, we believe it is valuable to include key elements of the proof techniques in the main body of the paper. We will revise Section 5 to provide additional intuition behind the results, as well as to highlight their implications and significance.
>
> > 4. Another rule-of-thumb is that all results proved in a paper, basically constitute its "Main Results". In the way that the paper is presently written, I feel that Proposition 3.6, which is a theoretical result of yours, remains "hidden" in the background Sec. 3. Could you reorganize things a bit, so that Proposition 3.6 is placed in the "Main Results" Sec. 4, where it belongs? The same holds for Definition 3.7.
>
> Proposition 3.6 is placed before the main results section because the derivation of the naive bound depends on it (specifically inequality (8a)). Moreover, variants of this result have appeared in prior work (as noted at the beginning of Section 3), and it does not constitute a fundamentally new contribution of our paper. That said, we agree that it would be more natural to include it alongside the main results in Section 5. In the camera-ready version we will restructure the paper accordingly to consolidate the primary results and improve clarity and readability.

---

> > ### Author Rebuttal · Reviewer_g2ws · 2026-04-03
> >
> > Most of my concerns have been addressed, I thank the authors. However, I'm still a bit skeptic w.r.t. the following.
> >
> > I've seen that the authors have incorporated experiments in their responses to the rest of the reviewers, and will include them in the first revision opportunity of their manuscript. Thus, I don't see why the following argument stands:
> >
> > > "However, due to space and time constraints, we are unable to include a detailed empirical evaluation in the current version."
> >
> > By all means, I don't suggest to compare with an arbitrary number of attacks found in relevant literature; but your aforesaid point does not constitute a clarification, as PGD is readily available in several benchmark and thus easy to include in your framework. Unless, of course, the authors have identified limitations, in which case it would be highly relevant to expose them in their response, so as to better position their paper.

---

> > > ### Author Response · Authors · 2026-04-04
> > >
> > > Than you for the follow up.
> > >
> > > We apologize for the omission. Our main concern regarding the implementation was that most publicly available PGD attack repositories are designed for classification tasks, whereas our experimental setting involves regression. We had not performed any evaluation prior to our rebuttal response.
> > >
> > > Below we present some preliminary results on adversarial robustness with respect to the width. In order to remain consistent with the presented theory, we follow the exact same training process described in Section 6.2. After non-adversarial training with SGD, we evaluate the robustness of the trained models by computing the mean adversarial loss
> > >
> > > $$\frac{1}{N}\sum_{i=1}^{N}\max_{\hat{x}\in \mathcal{S}i} \ell(f_{\theta_{s}}(\hat{x}), f_{\theta_{t}}(x_{i})),$$
> > >
> > > where we sample 512 inputs independently from the same distribution used for training, i.e., $x_{i} \sim \mathcal{U}([-50, 50])$, and $\mathcal{S}i := \overline{\mathcal{B}}(x_{i}, \epsilon)$ with a budget of $\epsilon = 5$. As suggested, we use the standard PGD method to compute the adversarial examples by setting the number of iterations to 40 and $\alpha = 0.25$.
> > > As the choice of  the teacher network can have a strong effect on the robustness independently of the width (and hence, we generate a fixed set of different teacher networks and for each of them, we learn it by training neural networks of different widths.
> > >
> > > The results (averages over five fixed randomly generated teacher networks) are summarized in the following table
> > >
> > > | Width | Avg Adversarial Loss |
> > > |------:|---------------------:|
> > > | 32    | 13.0444             |
> > > | 64    | 13.0096             |
> > > | 128   | 13.0142             |
> > > | 256   | 12.9832             |
> > > | 512   | 12.9984             |
> > > | 1024  | 13.0415             |
> > >
> > > These results align with the central theoretical implication of our work: as long as the trained networks remain in the lazy regime, increasing the width does not negatively impact the Lipschitz constant, and consequently, does not degrade the models’ adversarial robustness.
> > >
> > > In the camera-ready version, we will conduct a more thorough evaluation of adversarial robustness for the teacher–student regression experiment already considered. In addition, we will aim to include a small simulated classification task, as adversarial robustness is more naturally assessed in classification settings than in regression.

---

### Official Review · Reviewer_fYPx · 2026-03-11

**Soundness:** 3
**Presentation:** 3
**Significance:** 3
**Originality:** 3
**Overall Recommendation:** 5
**Confidence:** 5

**Summary:**

The paper derives theoretical bounds on the Lipschitz constants of neural networks with smooth activations with the He initialization and in the lazy training regime. The main result is that these Lipschitz these bounds are width-independent, and that lazy training implicitly regularizes the Lipschitz constant.

**Compliance With Llm Reviewing Policy:**

Affirmed.

**Final Justification:**

The rebuttal addressed my concerns and I believe that this is a good contribution.

**Key Questions For Authors:**

1. The main theorem is appealing in that it is width-independent, but the bound still appears to grow exponentially with depth. Could the authors discuss more explicitly whether the approach can provide a practically meaningful certificate for networks of moderate depth?
2. In Theorem 4.2, what concrete assumptions on the optimizer, step size, sample size, training time, or teacher-student model guarantee that the training trajectory remains inside a parameter ball of radius $R/sqrt(M)$? As written, the result seems to be conditional on lazy behavior rather than predictive of when lazy behavior occurs. In other words, how do we know if the neural network is over-parameterized and will enter lazy training regime in the first place?
3. Could the authors provide experiments on somewhat larger or more practically sized networks, ideally with at least one comparison to a certified Lipschitz estimator rather than only empirical estimates? This would help assess whether the theorem is mainly asymptotic and conceptual, or whether it also tracks robust behavior in realistic networks.
4. What is the main technical obstacle to extending the results to ReLU or piecewise-linear activations?
5. The related-work discussion would benefit from a clearer comparison to recent scalable certified Lipschitz estimation methods and local certified estimators, such as the below works:
 - Xu, Y. and Sivaranjani, S. ECLipsE: Efficient Compositional Lipschitz Constant Estimation for Deep Neural Networks. NeurIPS 2024.
 - Xu, Y. and Sivaranjani, S. ECLipsE-Gen-Local: Efficient Compositional Local Lipschitz Estimates for Deep Neural Networks. https://arxiv.org/abs/2510.05261
 - Huang, Y., Zhang, H., Shi, Y., Kolter, J. Z., and Anandkumar, A. Training Certifiably Robust Neural Networks with Efficient Local Lipschitz Bounds. NeurIPS 2021.
 - Shi, Z., Wang, Y., Zhang, H., Kolter, J. Z., and Hsieh, C.-J. Efficiently Computing Local Lipschitz Constants of Neural Networks via Bound Propagation. NeurIPS 2022.
6. Could the authors define exactly what notion of “local Lipschitz constant “is used throughout the paper? In some verification literature, "local" refers to a neighborhood around a single input point, whereas here the analysis appears to be over a bounded set.
7. Are Lipschitz estimates the only measure that is closely related to generalization? If not, what are the other options and would it be easier if we just switch to other measures instead of giving strict upper bound that is NP-hard?
8. How can the Lipschitz estimate benefit the training of over-parametrized models to improve robustness?

**Limitations:**

Yes

**Strengths And Weaknesses:**

Strengths:
1. The paper is overall rigorous, well-organized and easy to follow.
2. The paper derives a width-independent Lipschitz constant estimate for deep neural networks with He initialization, and it is especially promising for the large neural networks in use nowadays from a verification perspective.
3. Lipschitz upper bound in the lazy training regime is also useful as overparameterized neural networks are observed to remain close to their initialization and behave nearly linearly during training.

Weaknesses:
1. In the teacher–student experiment, the evidence of width independence is limited by the use of empirical Lipschitz constant rather than certified Lipschitz estimates and the neural networks are small (up to 3 layers). Consequently, it remains unclear how broadly the claim can hold.
2. In abstract and introduction, the connection between SGD training on overparameterized neural networks leading to interpolation of the training data and good generalization performance and the Lipschitz constant is not articulated clear enough, weakening the motivation to calculating the local Lipschitz constant of deep neural networks. It is recommended to explicitly mention the relationship here.
3. The theory excludes activation functions like ReLU, which is one of the most important activation classes in practice.
In addition, see key questions for authors below.

---

> ### Author Rebuttal · Authors · 2026-03-30
>
> We thank the reviewer for their constructive feedback and for the time and effort invested in evaluating our work. Below, we address the expressed concerns in detail.
>
> # Key Questions For Authors
>
> > 1. The main theorem is appealing in that...
>
> We recognize that the exponential dependence on depth in our bounds is the primary limitation of this work, a point we clearly emphasize in the Conclusion section. Nevertheless, our results provide meaningful robustness certificates for shallow and wide network architectures—settings that are often the sole focus of existing theoretical studies. At the same time, our findings suggest a broader robustness phenomenon that extends to deeper networks, even though the practical usefulness of our bounds diminishes as depth increases.
>
> > 2. In Theorem 4.2, what concrete assumptions...
>
> Indeed, the result in Theorem 4.2 is derived under the assumption of the lazy training regime. We note, however, that analyzing the training dynamics that give rise to this regime is beyond the scope of the present work. Nevertheless, this phenomenon has been extensively studied in the literature, and a substantial body of work characterizes the conditions under which lazy training emerges (e.g., see Loss landscapes and optimization in over-parameterized non-linear systems and neural networks, Theorem 8).
>
> > 3. Could the authors provide experiments...
>
> Thank you for the suggestion. In the camera-ready version of the paper we will include additional experiments to investigate the effect of depth on the Lipschitz constant, thereby accounting for larger network architectures. We also agree that a comparison with certified Lipschitz estimation methods is valuable. At the same time, however, it should be noted that the resulting estimates only provide upper bounds and need to capture the true growth rate. Below, we provide a basic (preliminary) comparison by using the ECLipsE_Fast method:
>
> | Width | 32 | 64 | 128 | 256 |
> |----------|----------|----------|----------|----------|
> | Naive  | $42.56$  | $97.44$  | $182.76$ | $367.07$ |
> | Certified  | $7.94$  | $12.65$  | $17.38$ | $24.40$ |
> | Empirical  | $0.45$  | $0.45$  | $0.37$ | $0.34$ |
>
> Although certified Lipschitz estimation methods provide tighter upper bounds than the naive approach, they still fail to capture the width independence observed at initialization.
>
> > 4. What is the main technical obstacle...
>
> To establish Theorems 4.1 and 4.2, we crucially rely on  Lemmas C.2 and C.3, which associate perturbations of inputs and weights, respectively, to perturbations of the Jacobian matrix of the network. Both of these lemmas require smoothness of the activation function which is violated by the ReLU or other piecewise-linear activations. To the best of our knowledge, most bounds  ReLU networks, in turn, are obtained by considering changes in the activation patterns during training, which is a substantially different method of analysis.
>
> > 5. The related-work discussion would benefit from...
>
> Thank you for bringing certified Lipschitz estimation to our attention. In the camera-ready version of the paper we will expand the related work section to provide a more detailed discussion of certified Lipschitz estimation methods, as they are directly relevant to our contributions.
>
> > 6. Could the authors define exactly what notion of “local Lipschitz constant “is used...
>
> Thank you for pointing, the potential for confusion in the definition of the local Lipschitz constant. In this work, we adopt the definition commonly used in the machine learning literature (see, e.g., Exactly Computing the Local Lipschitz Constant of ReLU Networks, Definition 1), where the local Lipschitz constant is defined over a set rather than at a single point, see our Definition 3.1. To avoid any ambiguity, we will include a brief clarification in the camera-ready version of the paper.
>
> > 7. Are Lipschitz estimates the only measure that is closely related to...
>
> There exist several classical complexity measures from statistical learning theory, such as the VC dimension and Rademacher complexity, that are directly linked to generalization. However, due to their inherently combinatorial nature, computing these quantities can be NP-hard even for relatively simple function classes. As a result, most works focus instead on deriving tractable upper bounds for these measures as well. In fact, an important direction of our current research is the development of novel bounds on the Rademacher complexity associated with the lazy training function class introduced in Definition 3.7.
>
> > 8. How can the Lipschitz estimate benefit the training of over-parametrized models to improve robustness?
>
> The provided estimates do not directly enhance robustness during training. Rather, they offer a partial explanation for a phenomenon widely observed in practice, namely, that also in the NTK regime, neural networks exhibit robustness both before and after training.

---

> > ### Author Rebuttal · Reviewer_fYPx · 2026-04-01
> >
> > The authors have answered all my questions. I believe that this is a good contribution and I have raised my score to a 5.

---

### Official Review · Reviewer_2Mt8 · 2026-03-13

**Soundness:** 3
**Presentation:** 3
**Significance:** 3
**Originality:** 3
**Overall Recommendation:** 4
**Confidence:** 3

**Summary:**

This paper investigates the Lipschitz continuity of overparameterized deep neural networks (DNNs) with smooth activation functions. The authors provide a theoretical gap where empirical evidence suggests the Lipschitz constant does not scale with network width, in contrast with existing theories often predict growth. Their results shows that these bounds are independent of the network width $m$. The theoretical results are supported by numerical experiments using a teacher-student framework.

**Compliance With Llm Reviewing Policy:**

Affirmed.

**Key Questions For Authors:**

1. Is He initialization critical for this analysis, or as long as the evolution is somewhat linear, the theoretical results will hold true?

2. Many existing works use explicit gradient penalties (such as $L_2$ on the Jacobian) to control the Lipschitz constant. What is the key difference or advantage of this approach against those?

**Limitations:**

The authors pointed out that "As a purely theoretical study, the direct societal impact is limited, though applications of Lipschitz-based robustness analyses in safety-critical settings may warrant careful consideration."

**Strengths And Weaknesses:**

I. **Strengths**

The work provides the width-independent Lipschitz bounds for a general class of smooth activations, beyond the existing literature that focuses primarily on ReLU networks. It extends the analysis from a static initialization point to a dynamic training regime (lazy training), showing that the smoothness properties persist during the early phases of learning. The presentation is clear and easy to follow.

II. **Weaknesses**

1. The "lazy training" regime is a specific optimization setting where weights do not move far from initialization. This only happens mostly for the overparametetrized network. Since the width is independent, it must grow the depth exponentially.

2. Some assumptions are made on smooth activation or bounded domain, but it is not a critical issue.

---

> ### Author Rebuttal · Authors · 2026-03-29
>
> We thank the reviewer for their constructive feedback and for the time and effort invested in evaluating our work. Below, we address the expressed concerns in detail.
>
> ## Weaknesses
>
> >  1. The "lazy training" regime is a specific optimization setting where weights do not move far from initialization. This only happens mostly for the overparameterized network. Since the width is independent, it must grow the depth exponentially.
>
> To tackle the case of overparametrized networks is exactly the main aim of the paper and is not possible if one applies previous results, which are not width-independent. Namely, the width-independence established by our bounds implies that one can overparameterize the network by increasing its width without adversely affecting its Lipschitz constant or robustness. In particular, increasing width does not degrade stability. This decouples overparameterization from robustness: width governs the optimization regime without inflating the Lipschitz constant, and no compensatory increase in depth is required.
>
> ## Key Questions for authors
>
> > 1. Is He initialization critical for this analysis, or as long as the evolution is somewhat linear, the theoretical results will hold true?
>
> We adopt this initialization scheme due to its widespread use in practice and in prior theoretical work. Nevertheless, we expect the width-independence of the Lipschitz constant to extend to a broader class of subgaussian weight initializations whose variance proxy scales inversely proportional to the square root of the network's width. In specific, the key requirement in our analysis is that the $\ell_2$ norms of the post-activations, $|| \mathbf{x}_{\ell} ||$, remain independent of the network width. We believe that any initialization scheme satisfying this condition would yield similar theoretical guarantees.
>
> > 2. Many existing works use explicit gradient penalties (such as  on the Jacobian) to control the Lipschitz constant. What is the key difference or advantage of this approach against those?
>
> In contrast to prior work that explicitly regularizes the Lipschitz constant, our results suggest that such regularization arises *implicitly* when the network operates in the lazy/NTK regime. In particular, even for wide, overparameterized networks, the Lipschitz constant remains controlled without requiring any modification to the training objective.

---

> > ### Author Rebuttal · Reviewer_2Mt8 · 2026-04-04
> >
> > I thank authors for their responses. The questions are all clear. Can authors further demonstrate their argument that
> >
> > "Namely, the width-independence established by our bounds implies that one can overparameterize the network by increasing its width without adversely affecting its Lipschitz constant or robustness. In particular, increasing width does not degrade stability."
> >
> > in Theorem 4.1 and 4.2? For example, from a network, if one increases its width, is that true that the width never enters and affects any constant in their bounds?

---

> > > ### Author Response · Authors · 2026-04-04
> > >
> > > Thank you for the follow up question.
> > >
> > > Indeed, the constants appearing in the bounds of Theorems 4.1 and 4.2 are universal, in the sense that they do not depend on any architectural or other parameters of the setup, including the network width. Consequently, increasing the width does not affect these constants, provided the network remains in the lazy regime. Please let us know if you have any specific concern about our proof of this width independence so we can provide additional explanations.
> > >
> > > This width-independence is also reflected in our experiments assessing the robustness to adversarial attacks that we performed on suggestion by reviewer g2ws: also empirically one does not observe any width dependence (see our response to their rebuttal acknowledgment).

---

### Decision · Program_Chairs · 2026-04-30

**Decision:**

Accept (regular)

**Comment:**

The paper presents Lipschitz bounds for deep neural networks. The main advantage of these bounds is being width independent, while also extending the bounds to the lazy training regime. The reviewers agree that this is an important contribution, and the paper is also well presented and easy to follow. They also appreciated the experimental results presented in the paper.

One reviewer raised concerns about the comparison to previous works and that the bound is exponential in the depth. I believe that this work extends previous results by eliminating the width independence, although in some cases this dependence is only logarithmic rather than polynomial. Also, having an exponential dependence on depth for Lipshcitz constant seems unavoidable.

I suggest that the authors incorporate the author's comments and add a clearer comparison to previous works, so the advantage of removing the dependence on the width is explicitly shown.